# Synthesis, Biosynthesis, and Biological Activity of Diels–Alder Adducts from *Morus* Genus: An Update

**DOI:** 10.3390/molecules27217580

**Published:** 2022-11-04

**Authors:** Carola Tortora, Luca Pisano, Valeria Vergine, Francesca Ghirga, Antonia Iazzetti, Andrea Calcaterra, Violeta Marković, Bruno Botta, Deborah Quaglio

**Affiliations:** 1Department of Chemistry and Technology of Drugs, Department of Excellence 2018–2022, Sapienza—University of Rome, P.le Aldo Moro 5, 00185 Rome, Italy; 2Dipartimento di Scienze Biotecnologiche di Base Cliniche Intensivologiche e Perioperatorie, Campus di Roma, Università Cattolica del Sacro Cuore, Largo Francesco Vito 1, 00168 Rome, Italy; 3Department of Chemistry, Faculty of Science, University of Kragujevac, R. Domanovića 12, 34000 Kragujevac, Serbia

**Keywords:** Diels–Alder type adducts, natural products, total synthesis

## Abstract

The plants of the Moraceae family are producers of a great variety of polyphenolic natural products. Among these, the Diels–Alder type adducts (DAAs) are endowed with a unique cyclohexene scaffold, since they are biosynthesized from [4+2] cycloaddition of different polyphenolic precursors such as chalcones and dehydroprenyl polyphenols. To date, more than 150 DAAs have been isolated and characterized from Moraceous and related plants. The main source of DAAs is the mulberry root bark, also known as “Sang-Bai-Pi” in Traditional Chinese Medicine, but they have also been isolated from root bark, stem barks, roots, stems or twigs, leaves, and callus cultures of Moraceous and other related plants. Since 1980, many biological activities of DAAs have been identified, including anti-HIV, antimicrobial, anti-inflammatory, and anticancer ones. For these reasons, natural DAAs have been intensively investigated, and a lot of efforts have been made to study their biosynthesis and to establish practical synthetic access. In this review, we summarized all the updated knowledge on biosynthesis, chemoenzymatic synthesis, racemic and enantioselective total synthesis, and biological activity of natural DAAs from Moraceous and related plants.

## 1. Introduction

Moraceae is a wide family of flowering plants, including more than 1400 species and around 60 genera, scattered worldwide. Among these, the mulberry tree, a widespread plant from the genus *Morus* cultivated in Asia (China, Japan) and Europe for long time, is commonly used as food for silkworms. On the other hand, the mulberry root bark is a crude drug known as “Sang-Bai-Pi” that is used in the Traditional Chinese Medicine as an anti-inflammatory, antipyretic, diuretic, expectorant, and laxative [1]. This plant material has been extensively investigated since the 1970s by researchers to isolate and identify the biologically active compounds. The extracts of *Morus* root bark are rich in polyphenols presenting a cyclohexene scaffold, known as Diels–Alder type adducts (DAAs), endowed with several biological activities, including anti-HIV, antimicrobial, anti-inflammatory, and anticancer activity [2]. It has been postulated that their biosynthesis involves an intermolecular [4+2] cycloaddition between a diene (a dehydroprenylphenol, usually a dehydroprenylflavonoid or a dehydroprenylstilbene) and a dienophile (a chalcone), catalyzed by a Diels–Alderase enzyme, since all isolated DAAs are optically active. To date, approximately 90 DAAs from Chinese *Morus* plants and more than 160 DAAs from Moraceae and related plants have been isolated, and their structure, source, optical rotation, relative configuration, and biological activities are summarized in a review [2]. All-*trans* or *cis*-*trans* isomers can be obtained, depending on the *endo-* or *exo*-adduct formed by cycloaddition, respectively. The absolute stereochemistry and optical rotation of DAAs have been established by Nomura et al. and will be discussed in Section 3.1. Some relevant DAAs that will be discussed in this review are reported in Figure 1.

Other types of optically active DAAs have been isolated from different species of the Moraceae family. Among these, brosimones A (**8**) and B (**7**) have been isolated from a Brazilian moraceous plant, namely *Brosimopsis oblongifolia*. Brosimone A (**8**) structure contains two cyclohexene units, suggesting a second intramolecular [4+2] cycloaddition in the biosynthetic pathway. Sorocenol B (**19**), a DAA isolated from *Sorocea Bonplandii*, contains a particular bicyclo [3.3.1] scaffold, which is supposed to be biosynthesized from a corresponding DAA by oxidative cyclization of the 2′ hydroxy group of the chalcone moiety [3]. Finally, ketalized DAAs are polycyclic compounds that derive from acid catalyzed ketalization of two correctly oriented phenol groups (only the *cis-trans* DAAs have the appropriate stereoelectronic requirements) on the carbonyl of the chalcone moiety (see Figure 2). Examples of ketalized DAAs are mulberrofurans F (**21**), G (**22**), K, and kuwanol A (**20**), isolated from *Morus* root bark. The first two are biogenetically related to mulberrofuran C (**3**) and chalcomoracin (**1**), respectively, by ketalization. Other structurally similar ketalized compounds, isolated from *Sorocea bonplandii*, are soroceal, soroceins A and B.

Nomura and co-workers comprised the first research group that intensively studied the biosynthesis of mulberry DAAs by means of isotopic labelling and feeding experiments on *Morus alba* cell cultures [1,4]. Since the 1980s to 2020, no further investigations were carried out to discover new insight into the biosynthesis of DAAs. Very recently, by using a pioneering new technique, the research groups of Gao and Lei identified, cloned, expressed, purified, and characterized the first stand-alone *M. alba* Diels–Alderase (MaDA). This enzyme, identified from *M. alba* cell cultures, is capable of catalysing the [4+2] cycloaddition of chalcones as morachalcone A and various natural and unnatural polyphenolic natural dienes [5]. The authors were also able to identify, clone, and express several other MaDAs with different substrate scope and *endo*/*exo* selectivity [6]. The identification of natural MaDAs represents significant evidence confirming the previously postulated existence of Diels–Alderase enzymes in the biosynthesis of DAAs. According to the literature, biomimetic [4+2] cycloaddition is considered as a key step for the total synthesis of DAAs [7,8,9,10,11,12,13]. From the first synthetic route established by Rizzacasa and co-workers via simple thermal conditions [7], many advances have been made, starting from classical methods for the Diels–Alder reaction, as Brønsted and Lewis acid catalysis, to modern methods such as electron-transfer initiation, silver nanoparticles catalysis, and chiral boron complex asymmetric catalysis [8,9,10,11,12]. Recently, a review reporting on the total synthesis of prenylflavonoid and related DAAs has been published by Lei and co-workers, but it is not updated, since it covers the literature through 2015 [13]. The present review, which covers the literature from 1980 to 2022, focuses on the updated biosynthesis, chemoenzymatic synthesis, total synthesis, and biological activity of DAAs from Moraceous plants. It is organized in chapters reporting the different topics analysed, including a deep analysis of stereoelectronic features and regioselectivity of the Diels–Alder reaction. Herein, we reported the total synthesis of DAAs from Moraceous plants updated through 2022, divided in racemic and enantioselective approaches. During the compilation of this review, a Chinese group published a review article presenting the isolation, structure, bioactivity, as well as the chemical and chemoenzymatic total syntheses of mulberry DAAs [14].

## 2. The [4+2] Cycloaddition Reaction as a Powerful Tool for Biomimetic Synthesis of DAAs

The Diels–Alder (DA) reaction is the most important example of a cycloaddition reaction, a very useful method for forming substituted new cyclohexene rings from alkenes and dienes [15]. The DA reaction has been widely applied over the last century in many total syntheses [16]. Its versatility allows for the construction of complex molecular architectures and has been exploited by both nature and synthetic chemists to construct a large variety of natural products. It is a concerted 4π_s_ + 2π_s_ cycloaddition, where the respective energy, geometry, and symmetry of frontier molecular orbitals (FMOs) of diene and dienophile play a crucial role in determining whether a process is allowed or forbidden (Figure 3a), namely, that it is energetically favourable and the regioselectivity and the stereochemistry of the newly formed ring [17,18,19,20].

In the transition state (TS), the diene adopts the *s*-*cis* conformation and approaches in approximately parallel planes of the substituted alkene, namely, the dienophile. The symmetry properties of the π-orbitals permit stabilizing interactions between “frontier” carbons of the diene and the dienophile. Usually, the strongest bonding interaction is between the highest occupied molecular orbital (HOMO) of the diene and the lowest unoccupied molecular orbital (LUMO) of the dienophile (Figure 3bII). Generally, the DA reaction is facilitated by electron withdrawing groups (EWGs) on the dienophile, which lower the energy of the LUMO. On the other hand, electron poor dienes react better with electron rich dienophiles (bearing electron releasing groups (ERGs)), that possess HOMO higher in energy and are accessible for diene’s LUMO, undergoing a so-called inverse electron demand DA (Figure 3bIII). According to FMO theory, it is also possible to understand and predict the regioselectivity of DA reactions when unsymmetrical dienes and dienophiles are involved (Figure 4).

Besides the regioselectivity, another feature related to unsymmetrical dienophiles is the two possible stereochemical orientations that the substituent can assume with respect to the diene: in the *endo* transition state, the substituent is oriented towards the π-system of the diene, while in the *exo* transition state, the substituent is oriented towards the opposite direction (Figure 5).

The *endo*-TS and, consequently, *endo*-product, is generally preferred when the substituent is an EWG, such as carbonyl, although it often leads to the more sterically hindered product [21]. This preference follows the “Alder rule”, and it is attributed to an interaction between π electrons of the diene and dienophile substituent via an alignment of the dipoles [22,23]; indeed, this preference is enhanced in polar solvents [24,25].

This review focuses on DAAs isolated from the mulberry tree and other plants of the genus *Morus*. These natural products, known as mulberry DAAs, result from the cycloaddition between 2′-OH-chalcone dienophiles and dehydroprenyl dienes, forming polyphenolic structures which are categorized according to the structure of the diene [26].

Synthetic studies supported by calculations reported by Boonsri et al. in 2012 highlighted the importance of the hydroxyl group at the C2′ of the chalcone moiety. Indeed, attempts of DA reactions to produce various mulberry DAAs starting from 2′-OMe chalcones (**25**) as dienophiles failed, while the experiments were successful using 2′-OH-chalcones (**24**) (Figure 6) [27]. This experimental outcome has also been computationally validated. The H-bond between the 2′-OH group and the carbonyl oxygen has different positive effects, thus accelerating the [4+2] cycloaddition: there is a stronger interaction between chalcone and diene due to the LUMO-lowering effect of the OH-carbonyl hydrogen bond. Density functional theory calculations showed that both *endo* and *exo* transition states starting from 2′-OH-chalcones have lower activation and reaction energies compared to 2′-OMe-chalcones, corresponding to 13–18-fold faster reactions under thermal reaction promotion; less energy is required to distort the diene to its TS geometry, therefore, the interaction between diene and dienophile is stronger due to the better coplanarity.

Selected total syntheses of mulberry DAAs disclosed in the next chapters have been reported since 2010. Besides the use of high temperatures to promote cycloaddition reactions, reported examples also employed Lewis acids (LAs) and Ag silica supported nanoparticles (AgNPs). A DA reaction, generally, can be catalysed by LAs that coordinate the dienophile and make it more electron-poor and, therefore, more reactive towards electron-rich dienes [28,29]. The use of LA, moreover, influences and often enhances the regio- and stereoselectivity. LA can also be employed together with chiral ligands, which are able to form rigid complexes with the dienophile and force the diene to approach selectively from one face of the double bond, resulting in cycloaddition products with high enantioselectivity. Particularly interesting are, from our point of view, the syntheses involving enantioselective approaches with respect to the cycloaddition step. Besides the first enantioselective synthesis of (−)-nicolaioidesin C, which employed (*R*)-(+)-camphor as a chiral auxiliary on the dienophile to induce asymmetry in a Brønsted acid catalysed cycloaddition [30], modern approaches take advantage of borane as LAs, in combination with chiral biaryl ligands, for both catalysing the cycloaddition and inducing asymmetry in product formation.

## 3. Biosynthesis of Mulberry DAAs

The mulberry DAAs’ biogenetic pathways have been investigated by Nomura et al. by incorporation studies of ^13^C-labelled acetates in *Morus Alba* cell cultures. Over a period of nine years, Nomura and co-workers were able to select some cell strains from *M. alba* callus tissues induced from the leaves or seedlings, with high productivity of pigments [1]. Six DAAs were isolated from pigments extracted from *M. alba* cell cultures, namely, kuwanons J, Q, R, and V, mulberrofuran E, and chalcomoracin along with their putative precursors, morachalcone A, isobavachalcone, and dehydroprenyl morachalcone A. By inspection of their chemical structures, Nomura and co-workers speculated that DAAs were biosynthesized from chalcones as dienophile precursors and dehydroprenylphenols as dienes. For example, chalcomoracin (**1**) derives from [4+2] cycloaddition of morachalcone A and dehydroprenylmoracin C [31,32,33,34]. The isolated DAAs and their putative precursors are reported in Table 1.

The subsequent studies confirmed the hypothesis that DAAs are biogenetically formed via a key Diels–Alder reaction between two phenols units. Depending on *exo*- or *endo*-addition, two different stereoisomers can be formed, namely all-*trans* or *cis-trans* adducts (See Figure 7).

The [4+2] cycloaddition is an intrinsic regioselective reaction, according to the HOMO and LUMO largest coefficient, which provides a more efficient overlapping. The regioselectivity in the biosynthesis of DAAs was proved by Nomura et al. by studying the Diels–Alder reaction between two model compounds (an unsubstituted chalcone **26** and 3-methyl-1-phenylbuta-1,3-diene (**27**)) and between the pyrolysis products of kuwanon G octamethyl ether (**31**): in both cases, only one regioisomer was formed (as a mixture of two stereoisomers, namely, all-*trans* and *cis-trans*) (Figure 8).

The absolute stereochemistry of DAAs has been investigated by means of X-ray crystallography and circular dichroism (CD) spectroscopy [4,35,36]. The absolute configuration of the three stereocenters in the cyclohexene moiety in the two types of stereoisomers of DAAs was established to be 3″R, 4″R, and 5″S for the all-*trans* adducts, while it was 3″S, 4″R, and 5″S for the *cis-trans* adducts (See Figure 7). The all-*trans* type adducts showed negative optical rotations, while the *cis*-*trans* adducts showed positive values [37]. A further confirmation of the involvement of a Diels–Alderase in the biosynthesis of DAAs was that all the isolated DAAs were optically pure (*ee* > 99%). It is worth mentioning the fact that both all-*trans* and *cis-trans* DAAs have been isolated from the same plant, suggesting that different isoforms of *endo*- or *exo*-selective Diels–Alderase enzymes should exist at the same time.

Another confirmation that DAAs are biosynthesized via [4+2] cycloaddition catalysed by Diels–Alderase enzymes in *M. alba* cell cultures came from administration experiments with *O*-methylated chalcones as modified precursors, since no *O*-methylated DAAs have been previously isolated from *Morus* plants or cell cultures. Administration of different *O*-methylchalcones resulted in the identification of several metabolites, including prenylated *O*-methylchalcones (which suggests that prenylation occurs after the aromatization of the chalcone skeleton) and prenylated *O*-methylated DAAs bearing the same stereochemistry of kuwanon J (**10**) and chalcomoracin (**1**), which indicates that the [4+2] cycloaddition involved in the biosynthesis of DAAs is enzymatic [38]. A further proof of the existence of an enzymatic system in *M. alba* cell cultures was achieved by the bioconversion of artocarpesin (**33**) into artonin I (**34**), a DAA biosynthesized from morachalcone A (**31**) as dienophile and dehydroprenylartocarpesin (**32**) as diene. By feeding artocarpesin (**33**) (a prenylflavone precursor never isolated from *M. alba* cell cultures) to *M. alba* cell cultures in the presence of endogenous morachalcone A (**31**), Hano et al. isolated artonin I (**34**) as a Diels–Alder product (Figure 9). This experiment confirmed that in the plant, an enzyme system is also present, capable of oxidizing the prenyl moiety of artocarpesin (**33**) to the dehydroprenyl group, the diene intermediate precursor of artonin I (**34**) [39].

### 3.1. Incorporation Studies with ^13^C-Labelled Acetate

The incorporation studies of [1-^13^C]-[2-^13^C] and [1,2-^13^C_2_] acetate were carried out by Nomura’s group on the *M. alba* callus cell cultures and the biosynthesis of some DAAs, namely, kuwanon J (**10**) and chalcomoracin (**1**). The ^13^C-labelling patterns of kuwanon J (**10**) and chalcomoracin (**1**), reported in Figure 10, showed high incorporation of acetate in their aromatic rings.

The labelling experiments highlighted that the biosynthetic pathways to kuwanon J (**10**) and chalcomoracin (**1**) share the same cinnamoylpolyketide precursor **35**. For the biosynthesis of chalcone moiety **36,** this intermediate undergoes a Claisen condensation followed by dehydration and aromatization. In contrast, the aldol condensation of the same precursor **35**, followed by decarboxylation and aromatization, leads to the 2-arylbenzofuran unit **38** (Figure 10) [5]. Three carbon atoms of prenyl units of chalcomoracin (**1**) (the unmodified one and the diene dehydroprenyl moiety deriving from diene precursor, which forms the cyclohexene ring in the DAA) also incorporate ^13^C-labelled acetate.

Prenylation of the DAAs precursor occurs through the reaction with dimethyl allyl pyrophosphate (DMAPP), or other sources of the prenyl unit, i.e., geranyl pyrophosphate (GPP) or farnesyl pyrophosphate (FPP), all deriving from mevalonate. As mentioned above in paragraph three, prenylation of DAA precursors in Moraceous plants usually occurs after aromatization of the polyketide precursors and is catalysed by a class of enzymes called prenyl transferases (PTs). The first regioselective PT from *M. Nigra* was isolated from a microsomal fraction of callus cell cultures and characterized by Vitali et al., 2004 [40]. The PT is a Mg^2+^ dependant microsomal enzyme able to transfer the prenyl group from DMAPP to the substrate, which showed selectivity for 2′,4′-dihydroxychalcones, as well for the isoflavone naringenin. Recently, several PTs have been isolated and characterized from some other Moraceous and related plants, including *M. alba* and *Cudrania tricuspidata*. These enzymes are able to transfer the prenyl moiety from DMAPP or GPP to 2′,4′-dihydroxychalcones or other flavones or isoflavones [41,42]. Experiments with ^13^C-labelled mevalonate or leucine showed no incorporation, probably because they are catabolized into CH_3_COSCoA, which then loses the labelling by elimination of CO_2_ in the tricarboxylic acid cycle. In any case, pulsed feeding experiments with [2-^13^C] acetate showed incorporation in the three carbon atoms of the prenyl tail (Figure 11). These findings suggested that the diene moiety of the dehydroprenyl-derivatives, precursors of the DAAs, is biosynthesized by oxidation of the prenyl moiety catalysed by an oxidase enzyme in Moraceous plants.

The shikimate pathway was also investigated by labelling experiments to confirm the involvement of the cinnamoylpolyketide intermediate **35** in the biosynthesis of DAAs. Administration of L-[3-^13^C]-phenylalanine and L-[3-^13^C]-tyrosine to *M. alba* cell cultures showed the incorporation into the β-carbon of chalcone moiety and 3-carbon of benzofuran moiety, confirming the involvement of the shikimate pathway leading to cinnamoyl or *p*-coumaroyl intermediates [37].

### 3.2. Discovery of the First Intermolecular Diels–Alderase from M. alba

Despite the advent of new techniques and the advancement of technology, no stand-alone intermolecular Diels–Alderase has been previously isolated. This is also due to the problems in plant enzyme identification, which arise from the limitations of transcriptomics-enabled methods and gene prediction based on specialized metabolism, which requires characterized enzymes as bait. In 2020, Gao et al. reported the identification and the functional and biochemical characterization of two enzymes from *M. alba* cell cultures, namely, the *M. alba* Diels–Alderase (MaDA) and the *M. alba* moracin C oxidase (MaMO) [43]. In preliminary assays, the diene precursor moracin C (**39**) and dienophile morachalcone A (**31**) were incubated with *M. alba* cell lysate, and chalcomoracin (**1**) was isolated as a major compound, along with a diene **40** derived from moracin C (**39**) oxidation. This evidence suggested that an oxidase (MaMO) and a [4+2] pericyclase (MaDA) are involved in the biosynthesis of chalcomoracin (**1**) (Figure 12).

The enzymes were identified and isolated through a new method, introduced by the authors, called biosynthetic intermediate probe (BIP)-based target identification. The approach is based on the use of a modified biosynthetic intermediate **41** endowed with a photoreactive group (a diazirine) and a tag (a propargyl group) and is capable of identifying unknown biosynthetic enzymes based on substrate binding (Figure 13a). The BIP is recognized from putative Diels–Alderase and transformed into the corresponding chalcomoracin derivative **42**. Irradiation with UV light activates the diazirine group, transforming it into a highly reactive carbene that covalently crosslinks with the target [4+2] pericyclase during the enzymatic reaction. The target protein is enriched by activity-based protein purification, which consists of ammonium sulfate fractionation, hydrophobic interaction chromatography (HILIC), ion exchange chromatography (IEC), and size exclusion chromatography (SEC). Identification of hit proteins by this method is highly reliable, since the undesired proteins can be easily removed. Then, the purified proteins were subjected to liquid chromatography–mass spectrometry/mass spectrometry (HPLC-MS/MS) proteomics analysis (Figure 13b), which revealed that they were berberine bridge enzyme (BBE)-like enzymes, usually involved in the biosynthesis of structurally similar dienes as intermediates of other natural products. Using the BIP method, a band corresponding to a molecular mass of about 60 kDa was obtained by UV irradiation. To identify the candidate gene pool and, thus, to permit further functional and biochemical analysis, this band, which corresponds to several BBE-like enzymes, has been subjected to the transcriptional analysis. Fourteen different transcripts were annotated, and the full-length complementary DNAs were cloned. Interestingly, the two cloned enzymes, the MaMO and the MaDA, share about a 50% identity with BBE-type tetrahydrocannabinolic acid (THCA) synthase.

Taking advantage of this approach, the authors were able to isolate, by MaDA, a flavin adenine dinucleotide (FAD)-dependent [4+2] pericyclase, which showed high catalytic efficiency and low substrate specificity, and the MaMO, a FAD-dependent oxidase that catalyses the oxidation of prenyl moiety of moracin C (**39**) to a reactive unstable diene. The MaMO and MaDA were successfully expressed in *Komagataella phaffii* and Hi5 insect cells, respectively, and purified. The obtained recombinant enzymes were fully characterized by measuring kinetic parameters such as k_M_ and k_cat_. Furthermore, DTF calculations were carried out along with some kinetic isotopic experiments (KIE) to estimate free energy variations for the DA transition state, the mechanism, and the intrinsic selectivity. The data supported a concerted asynchronous mechanism and the KIEs, together with DFT calculations, suggesting that the hydrogen bond between the carbonyl and the hydroxyl groups in the *ortho* position, contribute to the promotion of DA reaction by lowering the LUMO of the dienophile. Finally, full characterization of MaDA included crystal structure determination as FAD co-crystal, molecular dynamics (MD) simulations, and site directed mutagenesis.

## 4. Chemoenzymatic Total Syntheses of Natural and Unnatural Mulberry DAAs

The chemoenzymatic total synthesis is a well-established method for the preparation of natural products endowed with complex chemical scaffold, joining together the chemical synthesis for the preparation of advanced precursors that will be used for a late stage reaction, catalysed by enzymes which ensure high stereo- and enantioselectivity, leading to the formation of a single regioisomer with very high *ee* [44,45,46,47]. An intermolecular Diels–Alder reaction catalysed by newly discovered Diels–Alderases such as MaDA gave access to a new useful tool for the biotransformation of various dienes and dienophiles into natural or synthetic DAAs [45]. To explore the substrate scope and synthetic utility of the newly discovered MaDA, Gao et al. incubated some natural and synthetic dienes and dienophiles with the cloned enzyme MaDA [6]. The [4+2] cycloaddition reaction was promoted by dissolving substrates in Tris-HCl buffer in the presence of MaDA, at pH 8.0. An efficient chemoenzymatic total synthesis of guangsangon E (**43**), kuwanol E (**17**), kuwanon J (**10**), deoxyartonin I (**44**), and 18″-*O*-methyl chalcomoracin (**45**) was achieved through late-stage enzymatic reaction, and all the products were characterized by HPLC-MS, NMR spectroscopy, and polarimetry. All resulting compounds were optically pure (100% *ee*), and the spectroscopical data for synthetic compounds correspond to those reported for the natural compounds. According to the conversion of substrates into the products, the MaDA showed substrate selectivity for the dienophiles, while it was less selective for the tested dienes, which permitted the incorporation of different diene substituents in the mulberry DAA scaffold. In Figure 14, the tested substrates and DAAs natural product obtained from chemoenzymatic synthesis are reported [43].

The same approach was used for the total synthesis of artonin I (**34**) [48]. According to the retrosynthetic analysis reported in Figure 15, artonin I (**34**) can be obtained by a DA reaction between morachalcone A (**31**) and flavone diene **32**. This biotransformation could be catalysed by the same first intermolecular Diels–Alderase MaDA used for the chemoenzymatic total synthesis of the above-mentioned DAAs. The dehydroprenyl moiety can be introduced by Stille coupling from aryl iodide **46**. Iodide could be prepared from flavone **47** after the protection of hydroxyl groups and regioselective iodination.

The synthesis of diene **32** started from Claisen–Schmidt condensation of protected acetophenone **48** and benzaldehyde **49** in the presence of NaH, which supplied chalcone **50**. Flavone **51** was obtained from oxidative cyclization of chalcone **50** in the presence of iodine, followed by deprotection of benzyl groups by treatment with BBr_3_ and by in situ acetylation (Figure 16). Finally, the regioselective iodination in *ortho* to phenol moiety of flavone **52** with BTMA·ICl_2_, followed by Stille coupling with tributyl (3-methyl-1,3-butadienyl)stannane **54**, using Pd_2_(dba)_3_ and AsPh_3_ as a catalytic system, gave flavone diene **55**.

Diene **32** was obtained from deprotection of phenol groups by mild hydrolysis and then used in the enzymatic biotransformation. As expected, MaDA efficiently catalysed the Diels–Alder reaction between diene **32** and morachalcone A (**31**) as dienophile and showed excellent *endo*-selectivity, affording only the *endo*-isomer artonin I (**34**) in a 90% yield over two steps (Figure 17). The influence of pH and temperature on the enzyme activity was also investigated: the best conditions proved to be 50 °C and pH 8.5. The same experimental setup was used to synthesize dideoxyartonin I (**58**) in a 38% yield from the corresponding dideoxy flavone diene **57**; both products were obtained with *ee* > 99%.

The previously isolated MaDA showed only *endo*-selectivity, since it does not catalyse the DA reaction to DAAs with *exo*-configuration, such as mongolicin F (**2**), guangsangon J (**59**), and mulberrofuran J (**4**), while, among the *endo*-adducts, only the ones containing the prenyl group on the dienophile moiety were obtained, indicating a substrate specificity of the enzyme. These observations led us to postulate that other Diels–Alderase should exist in the same plant species, having diverse substrate scope and *endo/exo* selectivity [43]. Using MaDA and protein Basic Local Alignment Search Tool (pBLAST), 41 proteins were identified in the genome of *Morus notabilis*, 8 of which are closely related to MaDA, as suggested by phylogenetic analysis. Homologous genes of three putative Diels–Alderases from *M. notabilis*, namely MaDA-1, MaDA-2, and MaDa-3, were amplified from *M. notabilis* and expressed in High Five insect cells. The enzymatic Diels–Alder reaction between morachalcone A (**31**) or chalcone **24** and various natural dienes (prepared from chemical synthesis of acetylated precursors), such as **60**, **61**, **62**, and **63**, catalysed by MaDA-1, showed the same *endo*-selectivity as in the reaction catalysed by MaDA to produce chalcomoracin (**1**), mulberrofuran C (**3**), guangsangon E (**43**), kuwanon Y (**16**), and kuwanol E (**17**), while MaDA-2 and MaDA-3 were capable of producing *exo*-adducts, such as major products from the same precursors, as in mongolicin F (**2**), mulberrofuran J (**4**), albafuran C (**64**), guangsangon J (**59**), and *exo*-kuwanol E (*exo*-**17**) (Figure 18).

Furthermore, the MaDA enzymes also proved to catalyse the Diels–Alder reaction with high enantioselectivity, giving only enantiopure products (with *ee* > 98%) and high stereoselectivity. As the MaDA enzymes proved to be less selective for dienes, the substrate scope of the reaction was investigated to synthesize a library of unnatural DAAs [6]. It was found that the minimum structural requirement of diene for both MaDA-1 and MaDA-3 was the 2-dehydroprenylphenol moiety, as shown from computational calculations. The prenyl moiety is essential for the enzymatic activity; moreover, the substituents in the *para*- position can form stabilizing interactions with the enzyme, lowering the energy of the transition state. By [4+2] cycloaddition reaction between different natural and unnatural dienes and dienophiles, several unnatural DAAs (**65**-**67**) have been synthesized either as *endo*- or *exo*-isomers by using MaDA-1 or MaDA-3, respectively, as biocatalysts; their structures are reported in Figure 19.

To understand the mechanism of opposite stereoselectivities of MaDA enzymes, the crystal structure of the *exo*-selective MaDA-3 enzyme was solved. The combination of structure-based computational and mutagenesis studies revealed the key residues capable of stabilizing the *endo* or the *exo* transition state. According to the binding poses, the *endo*-transition state is stabilized by MaDA-1 or MaDA by a key hydrogen bond between R443 residue and the TS, while *exo*-selectivity is achieved by MaDA-3 through the formation of a cation–π interaction between R294 residue and TS. This pair of MaDA enzymes represent, thus, a powerful tool for the stereo- and enantioselective synthesis of natural or unnatural DAAs.

## 5. Total Synthesis of Mulberry DAAs

Due to the unique chemical scaffold and promising biological activities against HIV, tuberculosis, inflammation, and cancer [49,50,51,52,53,54,55,56,57], several research groups have developed, over the years, different approaches for the total synthesis of DAAs, including classical conditions for the Diels–Alder reaction such as thermal promotion, high pressure thermal catalysis, Brønsted acid catalysis, as well as modern methods such as electron-transfer initiation, silver nanoparticles catalysis, and chiral boron complex asymmetric catalysis [13].

### 5.1. Racemic Total Synthesis of Mulberry DAAs

#### 5.1.1. Chalcomoracin, Mongolicin F, Mulberrofurans C, and J Methyl Ethers

Chalcomoracin (**1**) and mulberrofuran C (**3**) were the first examples of natural products biosynthesized by an enzyme-controlled [58,59] intermolecular Diels–Alder reaction between a dehydroprenylphenol diene derived from an isoprenoid substituted phenolic compound and an alkene of a chalcone as the dienophile [26]. Total syntheses of chalcomoracin and mulberrofuran C heptamethyl ethers (**68** and **69**) were performed by Gunawan et al., 2010 [7], and the detailed mechanistic and theoretical study on the synthesis of these Diels–Alder adducts was provided by the same group in 2012 [27].

Retro-Diels–Alder reaction of chalcomoracin heptamethyl ether (**68**), given in Figure 20, is a result of its pyrolysis, affording highly unstable dehydroprenylbenzofuran **70** and chalcone **71**.

The proposed retrosynthetic approach towards the synthesis of mulberrofuran C heptamethyl ether (**69**) is depicted in Figure 21. The target compound could be obtained by a [4+2]-cycloaddition reaction between chalcone **25** and dehydroprenylbenzofuran **70**. The required diene **70** is presumed to be obtained using a Suzuki cross coupling [60] reaction of iodide **72** and corresponding boronate, and the benzofuran scaffold could be prepared by a selective Sonogashira coupling [61] between iodoalkyne **73** and iodide **74**.

First, synthesis of diene **70** was performed in accordance with Figure 22. Starting from readily available aldehyde **75**, after treatment with the Bestmann-Ohira reagent in basic conditions [62], alkyne **73** was obtained. Sonogashira coupling of **73** with the iodide **74** using Cs_2_CO_3_ as a base afforded the tolan **76** [63]. After its hydrolysis, the obtained phenol **77** was cyclized to benzofuran **72** using TBAF [64]. To further convert compound **72**, pinacol boronate **78** was prepared from enyne **79** by hydroboration [65,66] and via Suzuki coupling diene **70** was obtained in high yield.

The second reactant needed for the Diels–Alder cyclization proposed in Figure 21, chalcone **25**, was obtained by a Claisen-Schmidt condensation reaction between acetophenone **80** and aldehyde **81** (Figure 23). However, in every attempt to perform the final step, the reaction of chalcone **25** with diene **70** gave only traces of desired compound **69**.

Therefore, an alternative dienophile **24** was prepared by a Claisen–Schmidt condensation between phenol **82** and benzaldehyde **81** (Figure 24). Next, the heating of a solution of **24** and **70** in toluene for 16h afforded both the *exo*-product **83** (mulberrofuran J hexamethyl ether) and the desired *endo*-isomer **84** (mulberrofuran C hexamethyl ether) in a 1:1 ratio in 40% yield. Finally, methylation of *endo*-isomer **84** afforded the target compound mulberrofuran C heptamethyl ether (**69**). It can be concluded that the presence of the free phenol in chalcone **24** was crucial for the success of this Diels–Alder cyclization.

The synthesis of chalcomoracin heptamethyl ether (**68**) started with preparing the required prenylated chalcone **86** (Figure 25). For this reason, chalcone **24** was prenylated to ether **85**, which was then subjected to 1,3-shift mediated by Florisil^®^ in toluene [67], yielding the desired chalcone **86**.

The next step, a thermal Diels–Alder cycloaddition between diene **70** and chalcone **86,** was performed at 180 °C and afforded the desired chalcomoracin hexamethyl ether **88** and the additional product, mongolicin F hexamethyl ether **89** [68], in a 2:1 ratio (Figure 26). Finally, compound **88** was successfully methylated, affording the target compound chalcomoracin heptamethyl ether (**68**) [33].

#### 5.1.2. Kuwanon V, Dorsterone Methyl Ethers

In 2011, Rahman and co-workers reported the facile total synthesis of two type II DAAs, kuwanon V and dorsterone methyl ethers (**90** and **91**) [69]. The retrosynthetic plan includes a [4+2] biomimetic DA cycloaddition of chalcone diene **92** and chalcone **93** as a dienophile (Figure 27).

Diene **92** was synthesized starting from commercially available 2,4-dihydroxyacetophenone **94**, which was subjected to regioselective iodination with ICl to afford aryl iodide **95**. Methylation of compound **95** with (CH_3_)_2_SO_4_ followed by Claisen–Schmidt condensation with 4-methoxybenzaldehyde afforded iodinated chalcone **96**. The diene moiety was installed through Heck reaction of chalcone **96** with 2-methylbut-3-en-2-ol by using Pd(OAc)_2_, (*o*-tolyl)_3_P, and Et_3_N as a catalytic system. Allyl alcohol **97** was then easily dehydrated by treatment with acetyl chloride in the presence of pyridine to afford diene **92** (Figure 28).

On the other hand, dienophile **93** was prepared from previously known chalcone **98**; the 2-methoxy group was selectively deprotected with BCl_3_, then prenylation afforded *O*-prenylchalcone **100**, which was subjected to a 1,3-shift promoted by treatment with montmorillonite K10 to give *C*-prenylated chalcone **93** (Figure 29).

Finally, the Diels–Alder reaction between chalcones **92** and **93** was attempted under several conditions; first, the thermal promotion in a sealed tube at 160 °C afforded *endo*- and *exo*-adducts in a ratio of 3:2 and 55% yield (Figure 30). Different catalytic systems were tested, including the Porco’s catalyst (Zn_2_/Bu_4_NBH_4_, CoI_2_/1,10-phenanthroline), which proved to be efficient in promoting single electron transfer-initiated formal Diels–Alder cycloaddition, but the yield and *endo*/*exo* selectivity were comparable to the uncatalysed reaction. The best results were obtained by employing AgOTf/Bu_4_NBH_4_: kuwanon V (*endo*-adduct) (**90**) and dorsterone (*exo*-adduct) (**91**) pentamethyl ethers were obtained in 65% yield and 6:4 *endo*/*exo* ratio. Any attempt of deprotection of the hydroxyl groups with BCl_3_ resulted in decomposition.

#### 5.1.3. Sorocenol B

Sorocenol B (**19**) is a natural product found and isolated for the first time from the root bark of *Sorocea bonplandii* [3]. The structure of this molecule stands out by the presence of a bicyclo [3.3.1] core, which is also included in the structures of mulberrofuran I [70], australisin B [56], and mongolicin C [68]. The first total synthesis of (±)-sorocenol B (**19**) was performed by Cong et al., applying, for the first time, silver nanoparticle (AgNP)-catalysed Diels–Alder cycloadditions of 2′-hydroxychalcones [71]. The retrosynthetic analysis, presented in Figure 31, anticipates the formation of the target compound (±)-**19** by a biomimetic oxidative cyclization of *endo* (**102**) and/or *exo* (**103**) cycloadducts. The synthesis of these cycloadducts is presumed to be performed via AgNP-catalysed Diels–Alder cycloaddition of 2′-hydroxychalcone **104** and diene **105** [72], derived from readily available starting compounds chromene **106** and resorcinol **107**, respectively.

The synthetic procedure for preparing acetylated 2′-hydroxychalcone **104** is presented in Figure 32. First, chalcone **109** was obtained in excellent yield by a Claisen–Schmidt condensation reaction of chromene **106** and benzaldehyde **108** in the presence of NaH as a base. Afterwards, chalcone **104** was obtained in good yield in two steps, by MOM hydrolysis on chalcone **108** and subsequent acetylation [73].

The synthesis of required diene **105** (Figure 32) was achieved in four steps. First, hydroxyl groups of resorcinol **107** were protected using MOMCl and NaH, providing compound **110**, which was then subjected to a regioselective formylation towards benzaldehyde **111** [74]. Next, compound **111** was employed in an aldol condensation reaction with acetone and subsequent Wittig olefination of ketone **112**, affording diene **105**.

The reaction between acetylated 2′-hydroxychalcone **104** and diene **105** (Figure 33) presents the key Diels–Alder cycloaddition, which employs silica-supported silver nanoparticles (AgNPs) as a highly efficient and user-friendly catalyst (0.1 mol % of Ag loading). In this way, a mixture of *endo*/*exo* stereoisomers (**113** and **114**) was obtained with a 2:1 ratio. Since it was not possible to separate the *endo*-cycloadduct from the starting material, it was necessary to achieve the full conversion of chalcone **104**. It is worth noting that this conversion could not be achieved by the conventional thermal Diels–Alder reaction between **104** and **105**.

The next step involved the construction of a bicyclo [3.3.1] core (Figure 34), starting with the removal of acetyl-protective groups, giving **102** (*endo*) and **103** (*exo*) diastereoisomers. To achieve oxidative cyclization of the obtained intermediates, different oxidative conditions were explored (e.g., DDQ [75,76], CAN [77], and Pd(OAc)_2_/1,4-benzoquinone [78]), without success. The conversion of **102** (Figure 34a) was successfully achieved using Stoltz’s conditions for oxidative Wacker cyclization (catalytic Pd(OAc)_2_/pyridine in toluene under an oxygen atmosphere) [79,80,81], resulting in a bicyclic product **101** and its C-4 epimer **115**. The relative stereochemistry of both **101** and **115** were unambiguously determined using NOE analysis (Figure 35). Unfortunately, conversion of the *exo* diastereoisomer **114** under the same reaction conditions did not supply the desired product (Figure 34b). This can be explained via the proposed mechanism for the Pd(II)-mediated oxidative cyclization (Figure 34a), which indicates that in the case of **114**, *syn*-β-hydride is not feasible due to stereochemical restrictions.

In the final step, MOM hydrolysis of **101** in acidic conditions was performed without epimerization at the C-4 position, providing the target compound (±)-**19** in 74% yield. The relative stereochemistry of the obtained (±)-sorocenol B was confirmed by key NOE signals (Figure 36).

#### 5.1.4. Brosimones A and B

Qi et al. reported, in 2013, the first biomimetic total synthesis of brosimones A and B (**8** and **7**) through a multicatalytic dehydrogenative Diels–Alder (DHDA) cycloaddition of 2′-hydroxy-4′prenylchalcones [82]. The group reported for the first time, in 2010, the use of silica-supported silver nanoparticles (AgNPs) as a highly efficient catalyst for [4+2] cycloadditions of 2′-hydroxychalcones [72,83], so they exploited this catalytic system towards the biomimetic construction of Diels–Alder adducts natural products. Tandem reactions for the preparation of appropriate substrates have been planned, including, first, a dehydrogenation of prenyl chain to provide the diene in situ, followed by its AgNPs-promoted Diels–Alder cycloaddition to chalcone dienophile (Figure 37).

Preliminary studies on a model reaction set up the best combination of catalyst and oxidant to be a mixture of Pt/C and silica supported AgNPs, which also promotes the conversion of *endo*- to the desired *exo*-product, in an ambient air atmosphere, including cyclopentene acting as hydrogen scavenger. Subsequently, chalcone **118**, opportunely protected on the hydroxyl groups with Bn, has been dimerized using the combination of Pt/C-AgNPs to afford the cycloadduct **119** and **120** in 64% yield and *exo*/*endo* ratio of 1.2:1. Hydrogenolysis of benzyl protecting groups, with ammonium formate as an additive, provided brosimone B (**7**) (Figure 38). X-ray analysis of the corresponding methyl derivative confirmed the structure of the desired product.

Applying the same reaction conditions to *exo*-**120** did not afford the second DHDA to access the brosimone A derivative *exo*–*exo*-**121**. Replacing Pt/C with an excess of 2,3-dichloro-5,6-dicyano-1,4-benzoquinone (DDQ), in association with AgNPs, an *exo*/*endo*-**122** product was obtained, together with DDQ adduct **123**, in 17% and 34% yield, respectively. Nonetheless, by increasing the reaction temperature from 90 to 130 °C, it was possible to obtain the target adduct *exo*/*exo*-**121** in a 62% yield. Hydrogenolysis of the adduct provided brosimone A (**8**) in a 91% yield.

Both undesired product *exo*/*endo*-**122** and DDQ adduct **123** could be converted into an *exo*/*exo*-**121** derivative by treatment with AgNPs at high temperatures. This evidence suggested the occurrence of tandem retro-Diels–Alder/Diels–Alder, which is also consistent with temperature-dependent stereoselectivity, leading to the *exo*–*endo* product under kinetic control and to the *exo*–*exo* product under thermodynamic control (Figure 39). Predicted reaction energies and calculated transition states support this thesis. The computational model suggested a one-electron substrate oxidation process, accompanied by deprotonation of the 2′-hydroxyl group and complexation with Ag^I^ radical.

#### 5.1.5. Morusalbanol A Pentamethyl Ether

Morusalbanol A (**18**) was first isolated from the bark of *Morus alba* in 2012 by Yue et al. [84]. The main structural characteristic of this natural compound is an oxabicyclic moiety with solely *S*,*R*,*S* configurations at the C3, C4, and C5 chiral centres. It was proposed that this core skeleton biosynthetically derives from the intramolecular cyclization of a *cis*–*trans* DAA intermediate, which originates from an enzyme-controlled Diels–Alder reaction between a dehydroprenyl diene and a chalcone dienophile (Figure 40).

The synthetic methodology for the Diels–Alder and the subsequent formation of the oxabicyclic core was developed by Tee et al., 2015 [85]. In this study, it was observed that the use of silver catalysts (AgOTf and AgBF_4_) for the Diels–Alder reaction employing model dienes and dienophiles resulted in a selective formation of *exo*-diastereoisomer in low yield, followed by the decomposition of the starting diene. Accordingly, thermal conditions for the formation of the Diels–Alder adduct proved to be the most efficient ones (Figure 41).

Towards the synthesis of morusalbanol A pentamethyl ether (**128**), it was first necessary to obtain the corresponding dehydroprenyl diene **129** and chalcone dienophile **130**. The requested diene **129** was synthesized by a similar procedure as described above, reported by Rahman et al., 2011, for the preparation of **90** and **91** [69]. Starting from commercially available 2,4,6-trihydroxybenzoic acid, a sequence of methylation, regioselective iodination [86], and Heck coupling reaction with 2-methylbut-3-en-2-ol, using Pd(OAc)_2_ as a catalyst in the presence of K_2_CO_3_ in DMF at a slightly elevated temperature, provided the allylic alcohol, which, by dehydration, furnished the desired diene **129** [87]. Similarly, the required dienophile **130** was prepared by mono-methylation of commercially available 2,4-dihydroxyacetophenone, followed by Claisen–Schmidt condensation with 2,4-dimethoxybenzaldehyde [71].

As stated above, the formation of the desired Diels–Alder *endo*-adduct **131** was achieved under thermal conditions in toluene at 135 °C for 24 h (Figure 42).

Finally, it was necessary to perform a selective cleavage of the *O*-methyl ether of **131** (Figure 43), which is achieved using MgI_2_ (10 euiv) in Et_2_O-THF (1:1). The subsequent C3–C21 bond rotation was followed by an intramolecular, stereocontrolled cyclization affording the targeted (±)-morusalbanol A pentamethyl ether **128**.

The relative stereochemistry of **128** was determined using ^1^H NMR and NOESY experiments, and the structure was confirmed by the single-crystal X-ray crystallography.

#### 5.1.6. Kuwanol E

The first total synthesis of (±)-kuwanol E (**17**) was reported by Iovine et al., 2016 [88]. The retrosynthetic approach toward its synthesis is presented in Figure 44. It was presumed that target compound **17** could be obtained by cleavage of the *O*-methyl ether groups from kuwanol E heptamethyl ether (**133**), which could be prepared by a Diels−Alder [4+2] cycloaddition of morachalcone A trimethyl ether **86**, as dienophile, and stilbene tetramethyl ether **134**, as diene.

The retrosynthetic approach to morachalcone A trimethyl ether **86** proposes its synthesis starting from the commercially available 2-hydroxy-4-methoxyacetophenone and 2,4-dimethoxybenzaldehyde, via a Claisen–Schmidt condensation followed by base catalysed *O*-prenylation of the free hydroxyl group and subsequent prenyl 1,3-shift promoted by Montmorillonite K10 [89], similarly to that reported by Gunawan et al., 2010 [7].

To synthesize diene **134**, benzyl bromides **135a** and **135b** had to be prepared (Figure 45) starting from commercially available 4-bromo-3,5-dihydroxybenzoic acid (**136**). First, **136** was converted into ester **137** and then treated with methyl iodide, giving **138a**, which was further subjected to Finkelstein iodination, affording **138b** [90]. Compounds **138a** and **138b** were further reduced to benzyl alcohols **139a** or **139b**, which were finally converted to bromides **135a** and **135b**.

Halides **140a** and **140b** were prepared by one-pot Arbuzov and Horner–Wadsworth–Emmons reactions with commercially available 2,4-dimethoxybenzaldehyde (**81**) (Figure 46) [91,92]. Subsequent Suzuki–Miyaura coupling [60] with boronate **78** [93], using Pd_2_(dba)_3_ and *S*-Phos as the catalysts, yielded diene **134**.

The Diels–Alder cycloaddition of dienophile **86** and diene **134** was successfully performed in the presence of BH_3_·THF as a Lewis acid catalyst to enhance the reactivity of 2′-hydroxychalcone **86**. The reaction resulted in a mixture of two Diels–Alder adduct diastereoisomers (±)-**142** and (±)-**143** with an *endo*/*exo* = 4:1 ratio.

Finally, *endo*-isomer **142** was subjected to the cleavage of the methoxy groups providing the target compound (±)-kuwanol E (**17**) (Figure 47). The NMR spectroscopic data for synthetic and naturally occurring **17** were in agreement with each other and with the literature data [94].

#### 5.1.7. Kuwanon G and H

In 2021, Luo et al. reported the total synthesis of kuwanons G and H (**11** and **12**), two DAAs derived from the condensation of a chalcone with a flavone diene, that differs only in the presence of a prenyl group in kuwanon H (**12**) [95]. According to the retrosynthetic plan reported in Figure 48, kuwanons G and H (**11** and **12**) could be obtained from the demethylation of methoxy groups of heptamethyl ether precursors **144** or **145**, which, in turn, could be prepared from Diels–Alder cycloaddition of diene **29** with 2′-hydroxychalcones **24** or **86** as dienophiles. Chalcones **24** and **86** could be synthesized, as already mentioned, by Claisen–Schmidt condensation of commercially available dimethoxybenzaldehyde **81** and 2-hydroxy-4-methoxy acetophenone **82**, followed by prenylation **86**. Diene moiety of **29** could be introduced by Suzuki–Miyaura coupling of flavone aryl iodide **146**, that can be prepared by prenylation of β-diketone **147** and intramolecular cyclization. **147** could be obtained by esterification of intermediate **148** with 2,4-dimethoxybenzoic acid **149** and subsequent Baker–Venkataraman rearrangement.

The synthesis commenced from Claisen–Schmidt condensation of suitable acetophenone and benzaldehyde, followed by *O*-prenylation and a montmorillonite 1,3-shift of prenyl moiety, according to the already established route described by Iovine et al. in the synthesis of kuwanol E (**17**) [88]. For the preparation of diene **29**, intermediate **148** was obtained by regioselective iodination with KI/KIO_3_ of 2-hydroxy-4,6-dimethoxyacetophenone. To afford the ketoester **150**, 2,4-dimethoxybenzoyl chloride was treated with compound **148** in pyridine, which then undergoes Baker–Venkataraman rearrangement in the presence of sodium hydroxide to supply β-diketone **147** (Figure 49).

Several attempts at α-prenylation of β-diketone **147** always afforded a mixture of the desired prenylated β-diketone **152**, along with C,O-diprenylated byproduct **151**, which could be converted into the desired β-diketone **152** by treating the mixture in acidic conditions. Then, cyclization and dehydration by sulfuric acid treatment in ethanol gave iodoflavone **146** and a cyclic adduct with ethanol **153**, which can be separated and converted into **146** by acidic treatment at room temperature. With iodoflavonoid **146** in hand, the authors tried different conditions for the Suzuki–Miyaura coupling with pinacolboronate **78**, which was prepared according to the procedure established by Iovine et al. [88]. The Suzuki–Miyaura coupling was optimized, and the best conditions included the use of (PCy_3_)_2_PdCl_2_ as the catalyst in toluene at 60 °C in the presence of K_3_PO_4_ as a base, which gave the desired diene **29** in 41% yield (Figure 50).

The key Diels–Alder reaction was attempted in thermal conditions at high pressure in a sealed tube in toluene at 180 °C (Figure 51). *Exo-* (all-*trans*) and *endo*- (*cis-trans*) adducts of both dienophiles **24** and **86** with diene **29** were isolated in 23% and 28% yield, respectively. Finally, *exo*-adducts heptamethyl ethers **144** and **145** were deprotected by treatment with BBr_3_ to afford racemic kuwanon G and H (**11** and **12**), respectively, according to the demethylation conditions reported for the total synthesis of kuwanol E (**17**) (Figure 51) [88]. The racemate of kuwanon G (**11**) was separated by chiral HPLC to obtain the two enantiomers, which showed opposite specific rotations, in accordance with that reported for the naturally-occurring kuwanon G (**11**). Racemic kuwanon H (**12**) was subjected to the same procedure, also, in this case, two enantiomers with opposite optical rotations were obtained.

### 5.2. Enantioselective Total Synthesis of Mulberry DAAs

#### 5.2.1. Brosimones A and B and Kuwanons I and J

After only one year, the total synthesis of brosimones A and B (**8** and **7**) has been reported by the group of Porco. Han et al. accomplished the first enantioselective total synthesis of these two Diels–Alder adducts, along with the enantioselective total synthesis of kuwanons I and J (**9** and **10**), starting from the common intermediate chalcone **154** (Figure 52) [10].

Chalcone **154** was afforded by prenylation of the free hydroxyl group, followed by a 1,5-shift reaction catalysed by Montmorillonite K10 towards *para*-prenylated chalcone **155**, which was the synthetic precursor for brosimones A and B, and a 1,3-shift towards *ortho*-prenylated chalcone **156**, the synthetic precursor for kuwanons I and J (**9** and **10**). Prenylated chalcone **163** was a direct precursor of the diene **158**, obtained by a sequence of Schenck ene reaction/reduction followed by a dehydration reaction (Figure 53). After an extensive evaluation of diverse protecting groups, acetate was selected, as it could be removed smoothly under mild basic conditions.

Initially, the enantioselective Diels–Alder cycloaddition was developed and optimized on model substrates. Moreover, this reaction on 2′-hydroxychalcones has not been reported before. Taking inspiration from the work of Kelly et al. and Snyder et al. [8,9], the authors tested different chiral boron complexes and obtained excellent results using (*S*)-VANOL and moderate yield and enantioselectivity with (*R*)-VAPOL. With optimized reaction conditions in hand, the *para*-prenylated dienophile **155** and diene **158** underwent a Diels–Alder cycloaddition reaction in the presence of (*S*)-VANOL as chiral ligand, affording *endo*-**164** and *exo*-**165** in 71% yield with a 1.2:1 ratio, with excellent *ee* values for both (98% *ee* for *endo*-**164**, 93% *ee* for *exo*-**165**). Removal of acetyl groups provided (−)-brosimone B (**7**) in a 70% yield Figure 54.

For the final Diels–Alder intramolecular cycloaddition towards brosimone A (**8**), many issues had to be overcome, such as the ring strain of the 12-membered ring of the target molecule, the steric hindrance encountered by the boron ligand, and the difficulty to simultaneously activate the dienophile while deactivating the diene. Nevertheless, adding a slight excess of boron complex, the three expected products, *endo*,*endo*-**166**, *exo*,*endo*-**167**, and *exo*,*exo*-**168**, have been achieved in 28, 20, and 13% yields, respectively. The latter, the desired configuration, has been obtained in 95% *ee*. Finally, global deprotection of *exo*,*exo*-**168** under mild basic conditions efficiently afforded the desired natural product (−)-brosimone A in a 70% yield (Figure 55).

As previously said, the synthesis of kuwanons I and J proceeded from the *ortho*-prenylated dienophile **162**, prepared via 1,3-shift of the prenyl chain catalysed by Montmorillonite K10, which allowed the target regioisomer to be obtained in 37% yield (Figure 53).

Reaction conditions have been optimized for the preparation of diene **160** starting from dienophile **156** via subsequent Schenck ene reaction/reduction and dehydration. It has been found that a visible-light-mediated regioselective Schenck ene reaction using [Ru(bpy)_3_Cl_2_·6H_2_O] and MeOH was the best combination to achieve the tertiary alcohol in 55% yield (Figure 56). Following dehydration reaction in the presence of SOCl_2_/DBU provided diene **160** resulted in a 75% yield.

As already discussed, a model reaction has been studied for the subsequent stereoselective Diels–Alder cycloaddition. Different boron complexes with chiral ligand have been evaluated and, employing an excess (2.5 equiv) of (*R*)-VANOL *endo*-**171**, were accessed in 80% yield, at a 1.2:1 *endo*/*exo* ratio, with excellent stereoselectivity (97% *ee*). Lower enantiomeric excess has been registered in the same condition for *exo*-**170**, but when (*S*)-8,8′-dimethyl-VANOL was used, the *ee* value of the *exo* product was significantly improved, from 60% to 84%. Finally, global deprotection of *exo*-**170** and *endo*-**171** under mild basic conditions efficiently furnished the desired natural products kuwanons I (**9**) and J (**10**), both in 70% yields (Figure 57).

#### 5.2.2. Kuwanon X, Kuwanon Y, and Kuwanol A

The first enantioselective total syntheses of (−)-kuwanon X (**15**), (+)-kuwanon Y (**16**), and (+)-kuwanol A (**20**) were reported by Gao et al., 2016 [12]. Taking advantage of their previous experience [10,11], they stereoselectively built the cyclohexene skeleton, common for Diels–Alder type adducts derived from chalcone and dehydroprenylstilbene moieties, through an asymmetric Diels–Alder reaction catalysed by chiral biaryl ligands/boron Lewis acid.

The kuwanons X (**15**) and Y (**16**) could be generated through asymmetric Diels−Alder cycloadditions between dienophile **172** and diene **173** (Figure 58). Dienophile **172** is prepared, as described in the literature, through the Claisen–Schmidt condensation of the corresponding acetophenone and benzaldehyde [96]. Diene **173** is prepared by introducing the dehydroprenyl chain by Pd-catalysed Suzuki cross coupling on the iodinated corresponding stilbene **176** [7,97], previously obtained over three steps by Horner−Wadsworth−Emmons reaction.

A screening of seven different chiral biaryl ligands highlighted an unprecedented *exo* selectivity for this cycloaddition (Figure 59). In particular, (*S*)-VAPOL catalysed the Diels–Alder reaction with a very high *exo* selectivity (*endo*/*exo* 1:13) and high enantioselectivity, so it was directly used to obtain acetylated kuwanon X (**179**) in 79% yield and 94% *ee*. This *exo* selectivity is, of course, a problem for the synthesis of kuwanon Y (**16**) and, consequently, for the synthesis of kuwanol A (**20**), which is derived from the *endo*-kuwanon Y (**16**) through a biomimetic acid catalysed ketalization. Among some VANOL derivatives, the best result was achieved using (*R*)-6,6′-dibromo-VANOL, which allowed acetylated kuwanon Y (**180**) to be prepared in 80% yield and 96% *ee*, with 1:3.5 *endo*/*exo* selectivity. The high enantioselectivity and *exo* selectivity can be explained by the two transition states reported in Figure 60. The chiral ligand blocks the α-face of the dienophile, and the diene could only approach the dienophile by its less hindered β-face. Due to the steric hindrance between the acetyl group and the phenyl group, the less hindered *exo* transition state is more favoured, leading, therefore, to the observed *exo* selectivity.

After removal of protecting acetyl groups, treating the Diels–Alder adducts, *exo*-**179** and *endo*-**180** with K_2_CO_3_ corresponding kuwanons were isolated in 67% and 49% yield, respectively.

Both products have been treated with sulfuric acid to catalyse the biomimetic intramolecular ketalization, but only *endo*-**16** formed a ketalized product, confirmed as kuwanol A (**20**).

Nevertheless, the target compound has been obtained with very low yield, mostly due to the aforementioned low *endo*-selectivity. To improve the selectivity towards the *endo*-product, acetyl groups have been replaced with MOM protecting groups, which have shown no predominant *endo*/*exo* stereoselectivity in literature (Figure 61) [69,71].

MOM-protected diene **181** was synthesised and used in the asymmetric Diels–Alder cycloaddition catalysed by (*R*)-VANOL/boron Lewis acid. Without losing the enantioselectivity (*endo*-product obtained in 90% *ee*, in comparison to previously achieved 94% *ee*) the *endo*/*exo* selectivity changed to 1:1.2. This reduced preference towards the *exo*-product is attributed to an anticipation of the transition state in the reaction pathway due to higher energy of HOMO in MOM-protected diene **181**, therefore, a less steric demand to overcome for the *endo* compared to the *exo* transition state.

Finally, single step deprotection/ketalization, catalysed by sulfuric acid in EtOH, afforded kuwanon A (**20**) in 17.6% yield and the same high enantioselectivity.

#### 5.2.3. Sanggenons C and O

The first asymmetric total syntheses of flavonoid DAAs sanggenons C and O were reported by the group of Porco in 2016, and it has been achieved by a stereodivergent Diels–Alder cycloaddition between protected dienophilic 2′-hydroxy-chalcone and flavonoid diene **183** [98]. The diene can be derived from sanggenon A (**184**) via isomerization of its chromene ring (Figure 62).

Sanggenon A (**184**) and its precursor sanggenol F (**185**) were also synthesised for the first time by the same group, and their synthesis is reported in the same paper. Preparation of **185** started by tetra-MOM-protection of morin (**188**), commercially available, and proceeded with the 5-allylation of the remaining free hydroxyl group. Removal of the MOM-group from C-3 followed by a second 3-allylation reaction and MOM- deprotection afforded compound **191**, which is the substrate for the subsequent double Claisen rearrangement, catalysed by Yb(OTf)_3_ in a mixture of DCM and HFIP. The hydrobenzofuro[3,2-*b*]chromenone (±)-**192** was then the key intermediate towards sanggenon F. Protection of the hydroxyl group with TBS followed by a cross coupling reaction catalysed by Grubbs’ 2nd generation catalyst provided the two prenyl chains, and final removal of the TBS groups easily afforded sanggenon F (**185**) in excellent yield. Finally, dehydrogenation with DDQ in THF leads to the second desired natural product, sanggenon A (**184**), in good yield (Figure 63).

Initially, the authors’ idea was to directly use sanggenon A (**184**) as a diene precursor for the construction of sanggenons C and O (**13** and **14**) via stereodivergent Diels–Alder cycloaddition with 2′-hydroxychalcone **172**. Nevertheless, this approach led to decomposition and no product formation. Therefore, chromene derivative **194**, a variant of sanggenon A, was prepared from (±)-**193** with DDQ in THF to serve as a diene precursor. The TBS-protected chromene **194** could generate the required diene **183** in situ via a retro 6*π*-electrocyclization followed by deprotonation/protonation, which results in a formal [1,7]-H shift. In the presence of acetylated 2′-hydroxychalcone **172** employing AgNPs, two *endo*-products were obtained, which yielded a mixture (4:1 d.r.) of (±)-sanggenons C and O (**13** and **14**) after treatment, sequentially with NaHCO_3_ and NEt_3_·3HF (Figure 64).

The two natural products have the same absolute configuration of the *endo*-formed cyclohexene ring, but they are epimers at both C-2 and C-3. To make this cycloaddition enantioselective, the authors envisioned, as mentioned above, a stereodivergent approach and evaluated different borate and axially chiral ligands on a model reaction with simplified diene and dienophile. Model Diels–Alder adducts have been produced in high yield and excellent stereoselectivity by using (*S*)-3,3′-dibromoBINOL and triphenylborate. A borate complex with the model dienophile has also been isolated and characterized by X-ray, confirming the spatial disposition and the corresponding face selectivity (Figure 65).

The optimized conditions have been directly applied to synthesize sanggenons C and O (**13** and **14**) enantioselectively. The stereodivergent [4+2] cycloaddition between racemic diene precursor (±)-**194** and dienophile **172** catalysed by catalytic amounts of B(OPh)_3_ and (*R*)-3,3′-dibromoBINOL, followed by subsequent acetate and silyl groups deprotections, yielded almost exclusively *endo*-products sanggenons C and O (**13** and **14**) in a 2:1 ratio and 98% and 93% *ee*, respectively (Figure 66).

Computational studies have been conducted on the interacting complex between dienophile **197** and both enantiomers of a simplified variant of the diene (TMS instead of TBS), engaging in Diels–Alder cycloaddition. Models A and B, shown in Figure 66 with the lowest energy conformer of diene **183**, highlighted that model A is favoured compared to the corresponding model B using (*R*)-3,3′-dibromoBINOL. The steric interactions between the prenyl on the dienophile and phenyl groups on the diene are responsible for the significantly increased energy in assemblies related to model B.

## 6. Biological Activity of Mulberry DAAs

### 6.1. Antioxidant Activity

Mulberry DAAs comprise, in their structure, a flavonoid unit and polyphenol groups, which are known to contribute to good antioxidant activity. Hence, many DAAs showed promising antioxidant activities, with more than 50% of malondialdehyde (MDA) formation inhibition at a concentration of 10 μM. For example, kuwanon Y (**16**) [2] exhibited potent antioxidant activity with inhibitory rates of 56% at a concentration of 1 μM. Kuwanon X (**15**) [99] also exerted good antioxidant potency, with an 81% inhibitory rate of MDA, at a concentration of 10 μM. Chalcomoracin was also found to be effective in scavenging the superoxide anion, with radical scavenging activity of 71% in terms of inhibition of blue formazan formation [100]. In the same study, chalcomoracin was also shown to inhibit lipid peroxidation by exerting a protective effect against the destructive UV irradiation by reducing the level of MDA to 25% of untreated value at a concentration of 100 μM.

### 6.2. Anti-Inflammatory Activity

Besides their antioxidative properties, several DAAs have also demonstrated potent anti-inflammatory activity. Sanggenon C (**13**) and kuwanons G and H (**11** and **12**) were found to significantly affect arachidonate metabolism in rat platelets by a dose-dependent inhibition of the formation of 12-hydroxy-5,8,10-heptadecatrienoic acid (HHT) and thromboxane B, by the cyclooxygenase route [101]. Sanggenons C (**13**) and O (**14**) were tested for their inhibitory effect on COX-1 and COX-2, showing IC_50_ values of 10–14 μM and 40–50 μM [102]. Virtual screening and the experimental results obtained in two previously cited articles have provided important information regarding the structural requirements for the biological activity. An important structural feature for the inhibitory effect on both COX isoenzymes is the connection of the chalcone pharmacophore with flavonoid moiety via a cyclohexene ring. Additionally, the presence of a 2′,4′-dihydroxy group in the B-ring seems to be beneficial for COX-1 inhibition, while the isoprenyl group should be placed at the C2 position of the flavonoid C-ring for a positive effect on COX-2 inhibition. Furthermore, sanggenon C has inhibited TNF-alpha-stimulated PMN-HSC adhesion and expression of VCAM-1 by suppressing the activation of nuclear transcription factor-kappa B (NF-kB) [103]. Kuwanon J 2,4,10″-trimethyl ether also strongly inhibited NF-kB activity with the IC_50_ value of 4.65 μM [55]. Kuwanon G (**11**) has shown a significant inhibition of IL-6 production in lung epithelial cells (A549) and NO production in lung macrophages [104].

### 6.3. Cytotoxic Activity

Chalcomoracin showed moderate cytotoxic activities against five human cancer cell lines (A549, Bel-7402, BGC 823, HCT-8, A2780), with IC_50_ values ranging from 5.5 to 7.0 μg/mL, as determined by MTT assay [56]. The Diels–Alder type flavanone sanggenon C (**13**) isolated from *Morus cathayana* displayed potent cytotoxicity against two human oral tumour cell lines (HSC-2 and HSG), with 50% cytotoxic concentration, CC_50_ = 0.018 mM against HSC-2 [105]. This compound exhibited lower activity against normal human gingival fibroblasts (HGF) with a tumour specificity ratio of 2.3, suggesting a specific cytotoxic activity against cancer cell lines rather than normal cells. Kuwanons G and H (**11** and **12**) were found to inhibit the specific binding of the gastrin-releasing peptide (GRP) to GRP-preferring receptors, with K*i* = 470 and 290 nM, respectively. Kuwanon Y was shown to inhibit protein kinase C (PKC) with an IC_50_ value of 15 μM [106]. Sanggenon C (**13**) inhibited tumour cellular proteasomal activity and cell viability via induction of cell cycle arrest and cell death, but also induced necrotic cell death in cancer cells, which could limit the clinical potential of Sanggenon C [107]. Sanggenons C and O (**13** and **14**) and kuwanon J were evaluated using luciferase reporter assay and were found to inhibit hypoxia-induced HIF-1α accumulation in a dose-dependent manner in human hepatocellular carcinoma cell-line Hep3B cells (IC_50_ = 1.03 μM, for sanggenon O (**14**)) [108]. In addition, it was determined that these compounds are also active against hypoxia-induced vascular endothelial growth factor (VEGF) secretion in Hep3B cells (IC_50_ = 2.08 μM, for sanggenon O (**14**)). Sanggenon C (**14**) induced apoptosis in HT-29 colon cancer cells via increased ROS generation and decreased NO production, which is associated with inhibition of iNOS expression and activation of the mitochondrial apoptosis pathway. In another study, sanggenon C was found to inhibit the proliferation of prostate cancer PC3 cells in a dose- and time-dependent manner with a 24-h IC_50_ of 18.76 µmol/L, possibly by activating caspase 3 and caspase 9 pathways [109]. The cytotoxicity of (±)-sorocenol B (**19**) was screened against a panel of 60 human cancer cell lines, and it was concluded that the most sensitive tumour cell lines include prostate cancer PC-3 (GI_50_ = 1.1 μM), melanoma LOX IMVI (GI_50_ = 1.4 μM), leukaemia MOLT-4 (GI_50_ = 1.4 μM), and colon cancer HCC-2998 (GI_50_ = 1.4 μM) [71].

### 6.4. Antimicrobial Activity

Chalcomoracin exhibited considerable antimicrobial activity (MIC 0.78 mug/mL) against MSSAs (strains FDA 209P and Smith) and MRSAs (strains K3 and ST 28) [51]. The potency of inhibitory activity of this compound against these strains was similar to that of vancomycin (MIC 0.39–1.56 mug/mL). Kuwanol E (**17**) is one of the most potent natural compound inhibitors of *Mycobacterium tuberculosis* protein tyrosine phosphatase B (K*_i_* = 1.6 ± 0.1 μM), and the first reported non-peptidic natural compound inhibitor of PtpB [53]. Kuwanon G (**11**) and sanggenon C (**13**) inhibited the growth of oral pathogenic bacteria such as *Streptococcus mutans*, *Streptococcus sobrinus*, *Streptococcus sanguis*, and *Porphyromonas gingivalis*. Transmission electron microscopy (TEM) of kuwanon G-treated cells demonstrated remarkable morphological damage of the cell wall and condensation of the cytoplasm [110]. Kuwanon G (**11**) caused 100% mortality of *Ichthyophthirius multifiliis* theronts at the concentration of 2 mg/L and possessed a median effective concentration (EC50) of 0.8 36 ±0.04 mg/L against the theronts. The median lethal concentrations (LC50) of kuwanon G (**11**) to *Ichthyophthirius multifiliis* in grass carp was 38.0 ± 0.82 mg/L, which was approximately 50 times the EC50 for killing theronts [111]. Sanggenons C and O (**13** and **14**) showed an antifungal activity against *Venturia inaequalis* with IC_50_ values of 17.7 and 34.3 µM, respectively [112].

### 6.5. Miscellaneous Biological Activities

Different biological activities of Sanggenon C have also been evaluated. It was found to stimulate osteoblastic proliferation and differentiation, inhibit osteoclastic formation and function in vitro, and reverse the bone loss of zebrafish caused by prednisone [113]. Sanggenon C (**13**) also displayed cytoprotective effects by suppressing the inflammatory response and ROS production provoked during hypoxia via signalling mechanisms involving the activation of AMPK and simultaneous inhibition of mTOR and FOXO3a [114]. Yet another study revealed that Sanggenon C could exert protective effects against cardiac hypertrophy and fibrosis in response to chronic pressure overload via suppression of the calcineurin/NFAT2 pathway [115]. Mulberrofuran C (**3**) [116] and kuwanon G (**11**) [117] have been shown to possess hypotensive activity with intravenous injection of 1 mg kg^−1^, causing significant hypotension in rabbits. Mulberrofuran C (**3**) has also turned out to be a potent multi-targeted agent for Alzheimer’s disease (AD), with an IC_50_ value for EeAChE 4.2 ± 0.1 μM [118]. Kuwanon H (**12**) showed positive activity on HIV with an EC50 value of 1.95 μg/mL [119].

## 7. Conclusions

Morus and related plants are a rich source of DAAs, a wide class of prenylated polyphenols that contains a cyclohexene moiety derived from [4+2] cycloaddition in the biosynthetic pathway, endowed with important biological activities. The pioneering studies of Nomura et al. extensively analysed the biosynthesis of mulberry DAAs and revealed many insights into the biosynthetic pathways: first, the existence of an enzymatic system capable of oxidizing the prenyl moiety of a polyphenol precursor as chalcones, flavonoids, stilbenes, etc. to a dehydroprenyl diene moiety. Moreover, the existence of another family of enzymatic systems, namely, the intermolecular Diels–Alderases that catalyse the [4+2] cycloaddition between chalcones as dienophiles and different dehydroprenyl polyphenolic precursors, such as dienes, was proved by ^13^C incorporation studies and by the isolation of enantiopure *endo-* (*cis-trans*) or *exo-* (all-*trans*) adducts. Despite the advent of new genetic engineering techniques and the advancement of technology, also due to the problems in plant enzymes identification, no stand-alone intermolecular Diels–Alderase has been previously isolated. Very recently, the first Falvin-dependent intermolecular Diels–Alderase from *Morus alba* cell cultures was isolated by Gao et al., namely, the Morus alba Dials-Alderase (MaDA). The enzyme was fully characterized, and some isoforms were cloned and produced with opposite *endo*/*exo* selectivity. Moreover, moracin C oxidase, an enzyme capable of dehydrogenating the prenyl moiety of moracin C to the dehydroprenyl diene group, has been identified and successfully cloned and expressed. The discoveries gave input to the exploitation of these new *endo-* or *exo*-selective Diels–Alderase enzymes in the production of many natural and unnatural DAAs by chemoenzymatic synthesis.

The use of mulberry root bark in Traditional Chinese Medicine for the treatment of various diseases has illuminated the importance of the isolation, identification, and biological screening of contained bioactive compounds. DAAs represent one of the most important classes of these compounds, showing prominent biological activities such as antioxidative, anti-inflammatory, cytotoxic, antimicrobial, and anti-HIV agents. However, the amount of each compound made available by the chemoenzymatic synthesis in the batch is limited and, thus, not sufficient for the preclinical development of a lead compound. In the future, flow chemistry systems with immobilised MaDA would provide higher quantities of DAAs. The chemical total synthesis represents a valid alternative. For this reason, many researchers have made a lot of effort to develop practical synthetic procedures for the obtaining of DAAs. It is worth mentioning that the key synthetic step for all the procedures is a biomimetic [4+2] cycloaddition, therefore, many attempts have been made to develop a highly stereo- and enantioselective version of the Diels–Alder reaction. Different studies revealed that the presence of an *ortho* hydroxyl group that forms an intramolecular hydrogen bond is fundamental in lowering the HOMO–LUMO gap, thus accelerating the reaction rate. The same effect was exploited by the coordination with several Lewis acids and silver nanoparticles. The first attempts of the Diels–Alder reaction for the synthesis of DAAs were thermal or high pressure, but the yield and *endo*/*exo* selectivity was not optimal. Then, new methods, including single electron transfer by Porco’s catalyst and the use of silver nanoparticles, were introduced to enhance stereoselectivity and yield. Finally, more efficient catalytic systems such as Brønsted acid and boron chiral ligands were introduced to increase the stereoselectivity and induce enantioselectivity in the DAA products. According to the synthetic procedures, which vary depending on the final synthetic target, some procedures are commonly adopted, such as the installation of the diene moiety (dehydroprenyl) through a Palladium catalysed cross coupling as Stille or Suzuki–Miyaura reactions or the introduction of the prenyl group through *O*-prenylation of a free *ortho* hydroxyl group, followed by a 1,3-shift of the prenyl group. It is noteworthy that, very often, such a kind of prenyl 1,3 shift is incorrectly reported as a [1,3]-sigmatropic rearrangement, but this reaction has a concerted pericyclic mechanism; thus, the expected products should bear a 1,1-dimethylallyl group, and such compounds have never been reported among the products. Moreover, the formation of a deprenylated by-product and the fact that the reaction is promoted by Lewis’ acids suggest that the reaction occurs as an ionic shift of the prenyl group as allylic carbocation. Finally, Brønsted acid and boron chiral ligands were investigated as efficient catalytic systems capable of inducing enantioselectivity in the DAAs. Many advances have been achieved, and today, several stereo- and enantioselective chemical synthetic procedures for the preparation of enantiopure unnatural and unnatural DAAs are available, even if the control of *endo*/*exo* stereoselectivity remains a problem to be addressed. The use of pure Diels–Alderase enzymes with specific *endo-* or *exo*-selectivity is the best way to solve the problem, and the enzyme also provides enantioselectivity, but the use of chemical synthesis remains a gold standard for the synthesis of DAAs in a large scale.

## Figures and Tables

**Figure 1 molecules-27-07580-f001:**
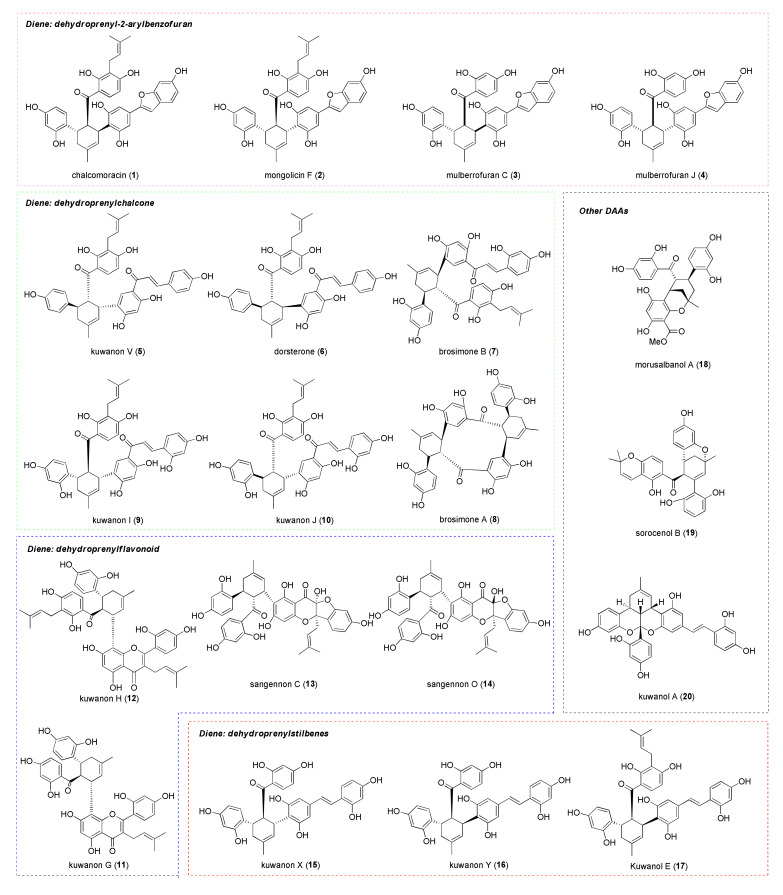
Diels–Alder adducts from the Moraceae family, whose synthetic approaches will be discussed in this review.

**Figure 2 molecules-27-07580-f002:**
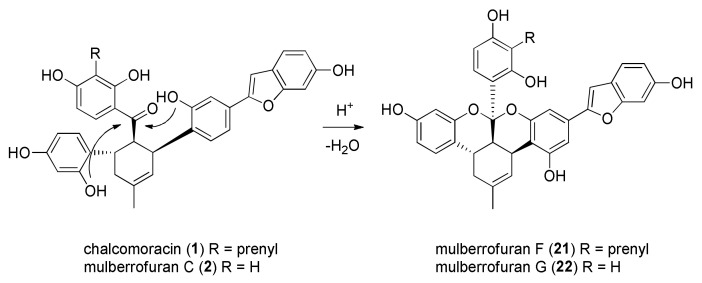
Formation of ketalized compounds from DAAs.

**Figure 3 molecules-27-07580-f003:**
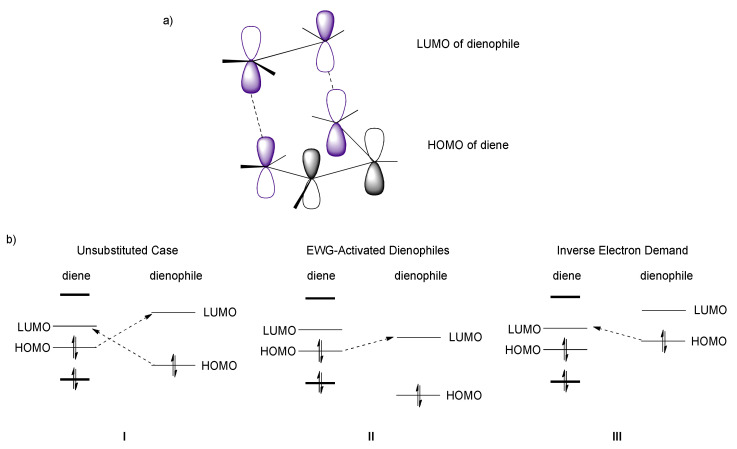
(**a**) Representation of HOMO and LUMO in the DA transition state and (**b**) types of FMO interactions. (**I**) standard HOMO–LUMO interaction, comparable energies; (**II**) HOMO of the diene interacts with LUMO of the dienophile; (**III**) LUMO of the diene interacts with HOMO of the dienophile.

**Figure 4 molecules-27-07580-f004:**
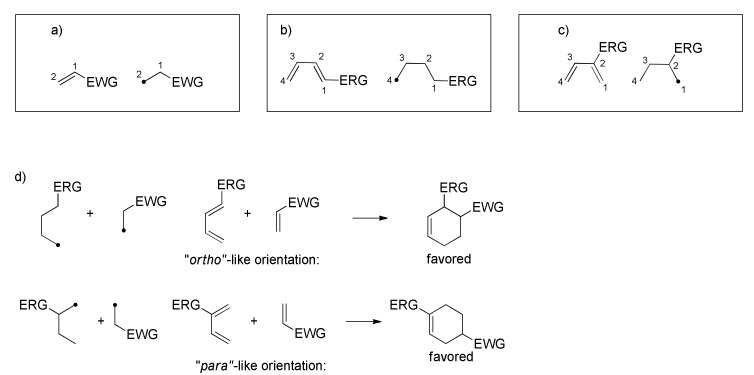
Regioselectivity of DA reactions according to FMO theory: (**a**) the carbon bonded to EWG on the dienophile has a higher coefficient than the other carbon; (**b**) the carbon C-1 directly bonded to ERG on the diene has a higher coefficient than others; (**c**) the carbon C-1, adjacent to the carbon bonded to ERG, has a higher coefficient than others; (**d**) “frontier” carbons with a higher coefficient match in the forming cyclohexene ring.

**Figure 5 molecules-27-07580-f005:**
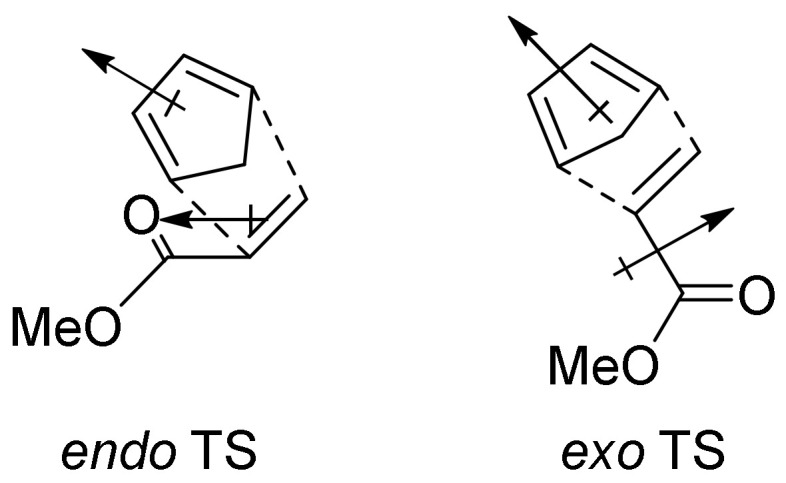
*Endo*- and *exo*-transition states.

**Figure 6 molecules-27-07580-f006:**
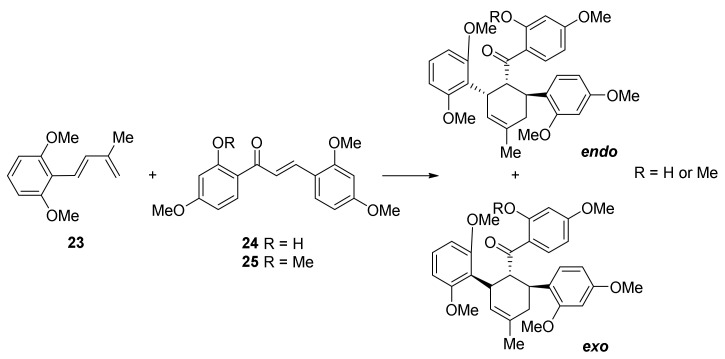
Model Diels–Alder reaction between 2′OH- (**24**) or 2′-OMe-chalcones (**25**) and diene (**23**) and corresponding endo and exo products.

**Figure 7 molecules-27-07580-f007:**
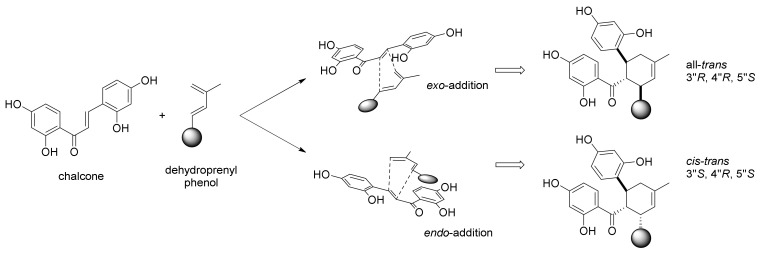
Key biosynthetic [4+2] cycloaddition and absolute configuration of all-*trans* and *cis-trans* DAAs.

**Figure 8 molecules-27-07580-f008:**
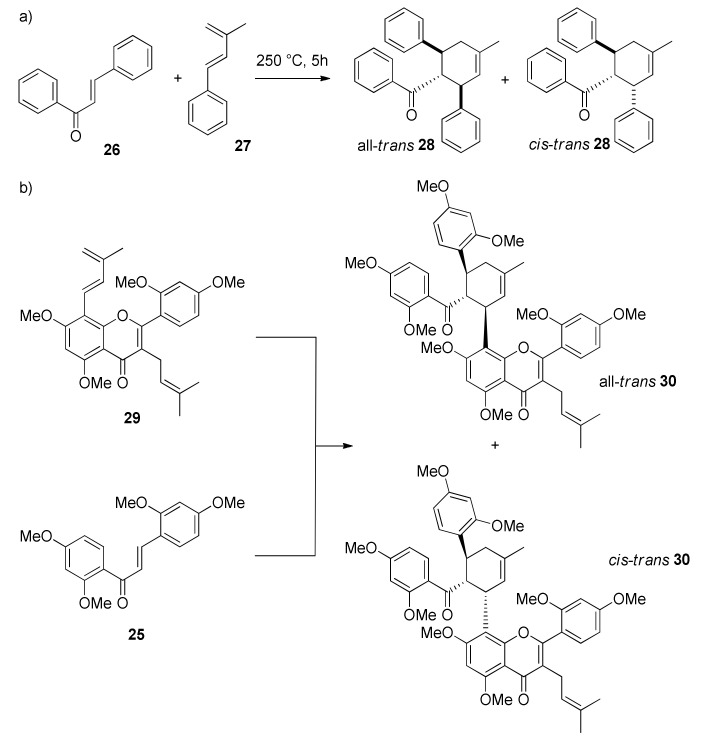
Diels–Alder reaction of (**a**) model chalcone **26** with dehydroprenyl benzene **27** and (**b**) pyrolysis products of kuwanon G octamethyl ether (**30**) affording endo- or *exo*-adducts of the same unique regioisomer.

**Figure 9 molecules-27-07580-f009:**
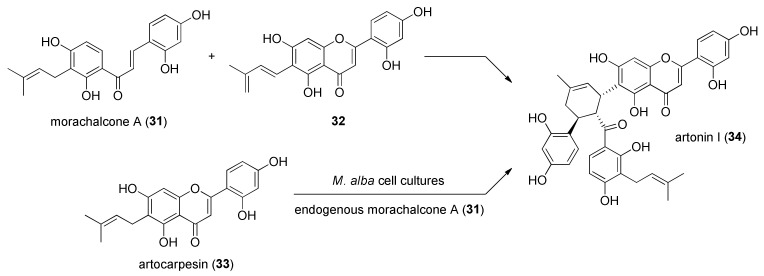
Bioconversion of artocarpesin (**33**) to artonin I (**34**), by feeding experiment with *M. alba* cell cultures.

**Figure 10 molecules-27-07580-f010:**
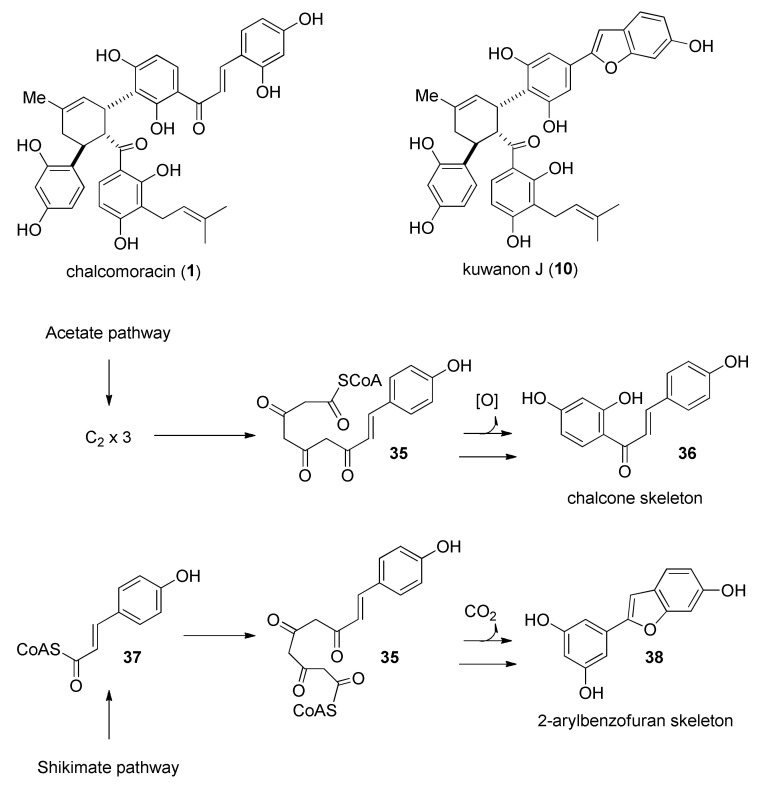
Biogenesis of chalcone **36** and 2-arylbenzofuran **38** in *M. alba* cell cultures producing kuwanon J (**10**) and chalcomoracin (**1**).

**Figure 11 molecules-27-07580-f011:**
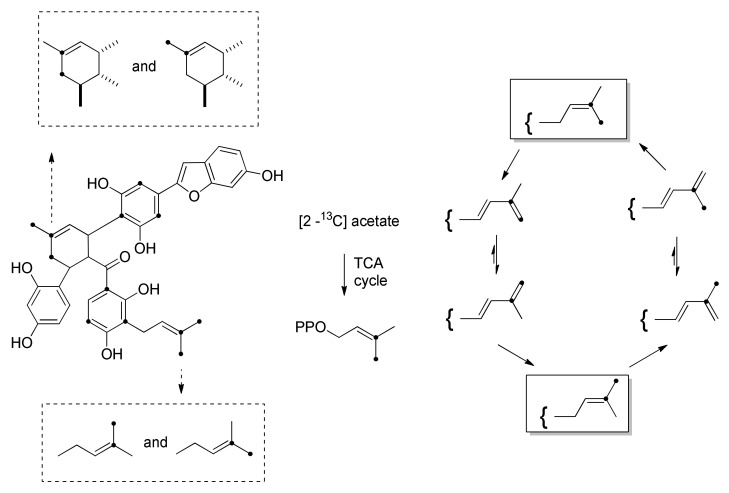
Pulsed administration experiment with [2-^13^C] acetate revealing the polyketide moieties and the oxidation of the prenyl group to the dehydroprenyl moiety.

**Figure 12 molecules-27-07580-f012:**
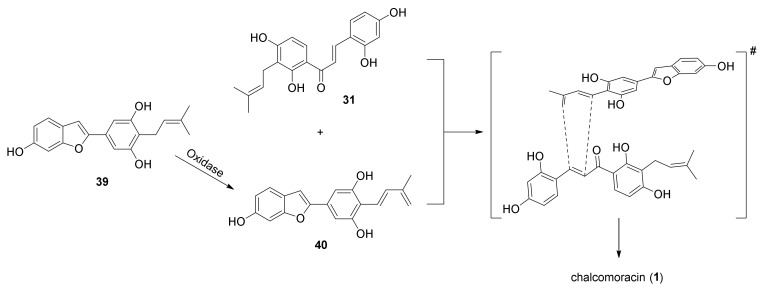
Proposed biosynthesis of chalcomoracin (**1**) from moracin C (**39**) and morachalcone A (**31**).

**Figure 13 molecules-27-07580-f013:**
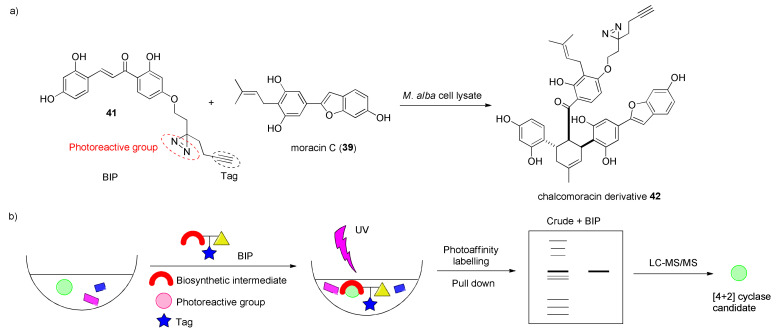
BIP-based identification method of putative [4+2] pericyclase from *Morus alba*.

**Figure 14 molecules-27-07580-f014:**
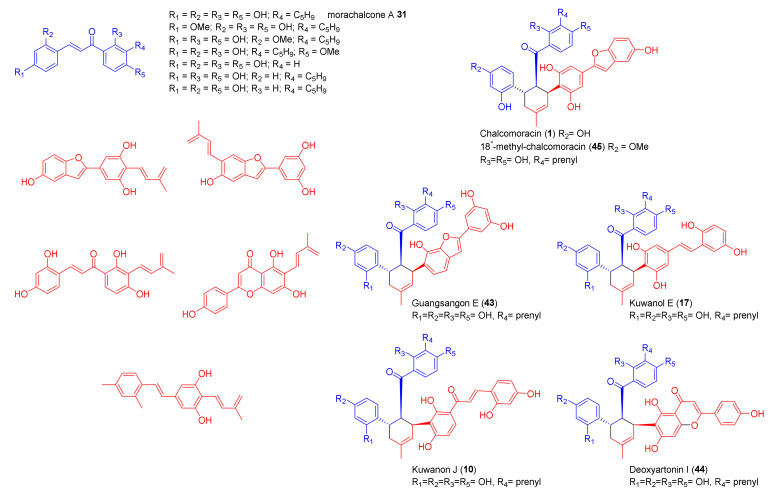
Substrate scope of MaDA and chemoenzymatic synthesis of natural DAAs.

**Figure 15 molecules-27-07580-f015:**
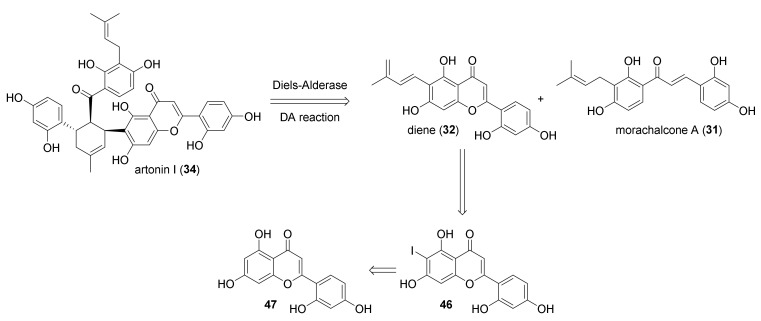
Retrosynthetic analysis of artonin I (**34**) by chemoenzymatic synthesis.

**Figure 16 molecules-27-07580-f016:**
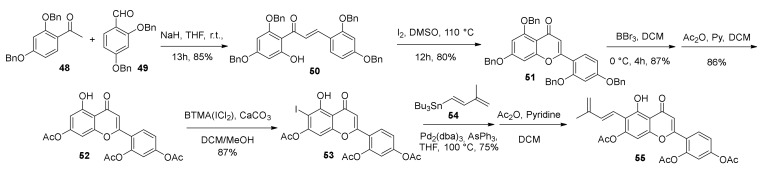
Chemical synthesis of key flavonoid diene **55**.

**Figure 17 molecules-27-07580-f017:**
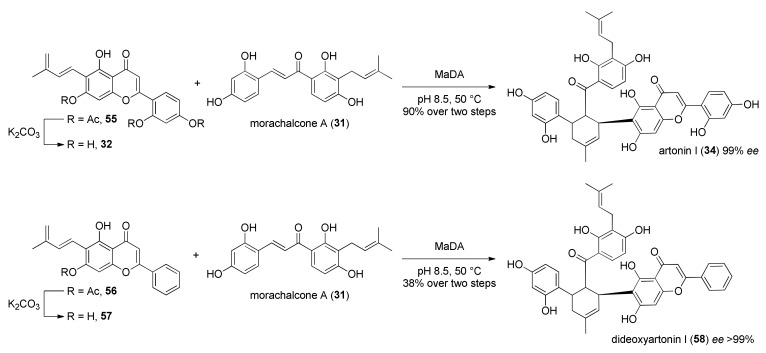
Chemoenzymatic total synthesis of artonin I (**34**) and dideoxyartonin I (**58**).

**Figure 18 molecules-27-07580-f018:**
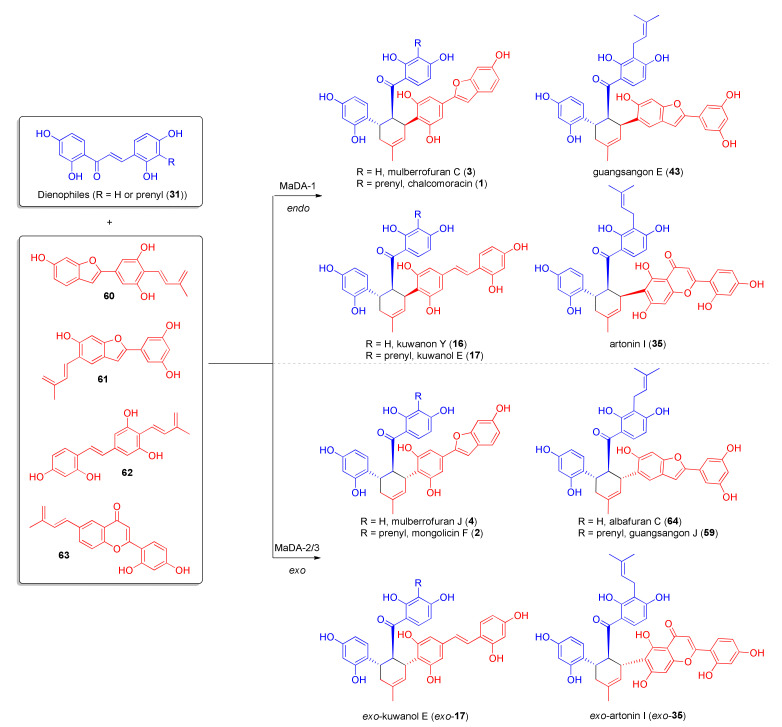
Chemoenzymatic total synthesis of natural DAAs using MaDA-1 (*endo*-selective) or MaDA-2/MaDA-3 (*exo*-selective).

**Figure 19 molecules-27-07580-f019:**
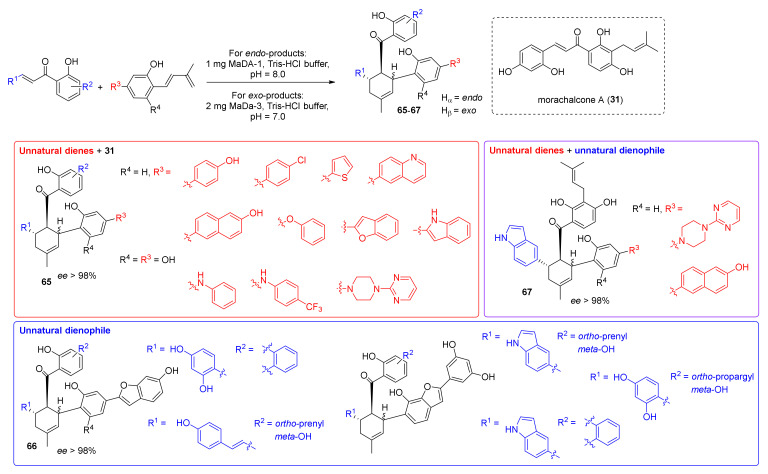
Chemoenzymatic synthesis of unnatural DAAs using MaDA-1 and MaDA-3.

**Figure 20 molecules-27-07580-f020:**
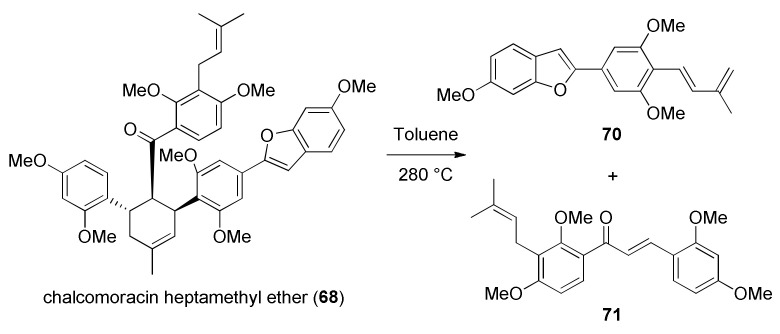
Retro-Diels–Alder reaction of chalcomoracin heptamethyl ether (**68**).

**Figure 21 molecules-27-07580-f021:**
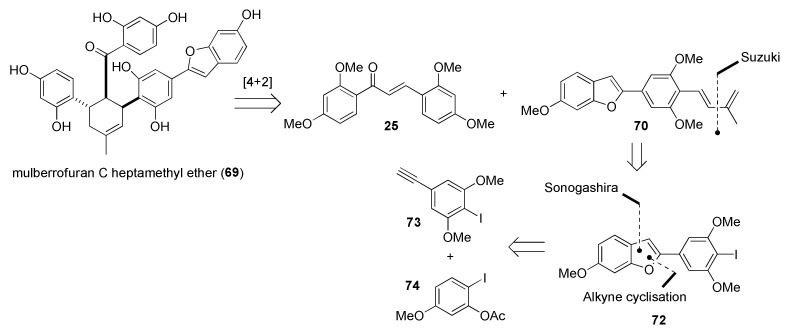
Retrosynthesis of mulberrofuran C heptamethyl ether (**69**).

**Figure 22 molecules-27-07580-f022:**
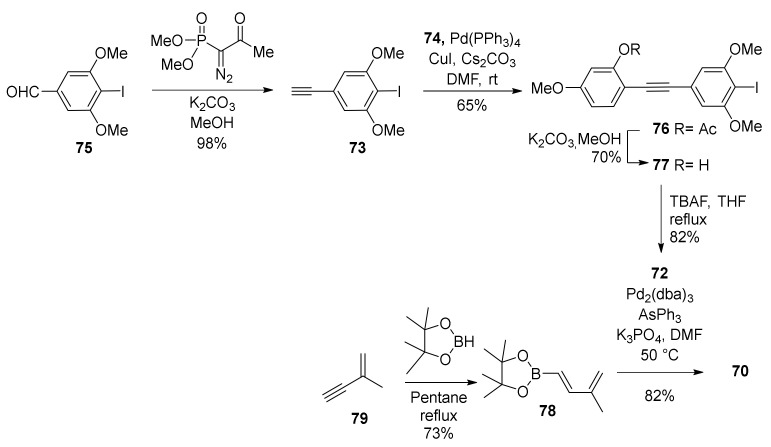
Synthesis of diene **70**.

**Figure 23 molecules-27-07580-f023:**
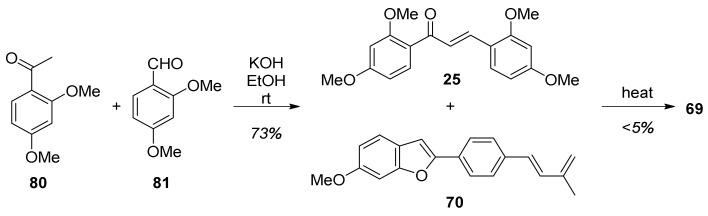
Synthesis of chalcone **25** and attempted Diels–Alder reaction between **70** and **25**.

**Figure 24 molecules-27-07580-f024:**
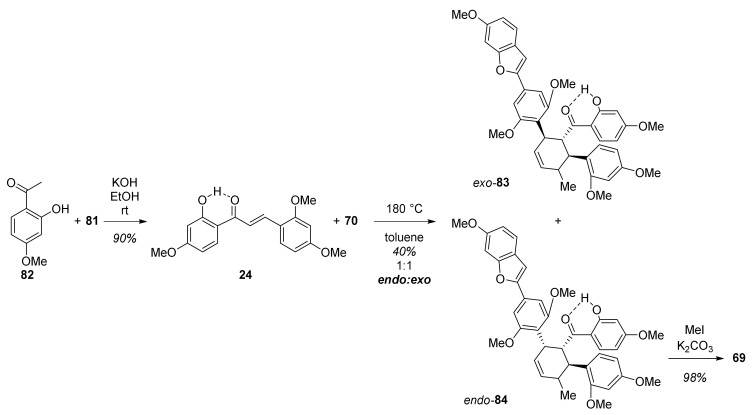
Diels–Alder reaction between **70** and phenolic chalcone **24**.

**Figure 25 molecules-27-07580-f025:**
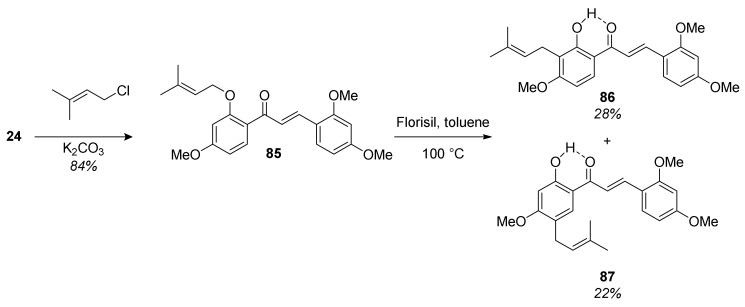
Synthesis of chalcone **86**.

**Figure 26 molecules-27-07580-f026:**
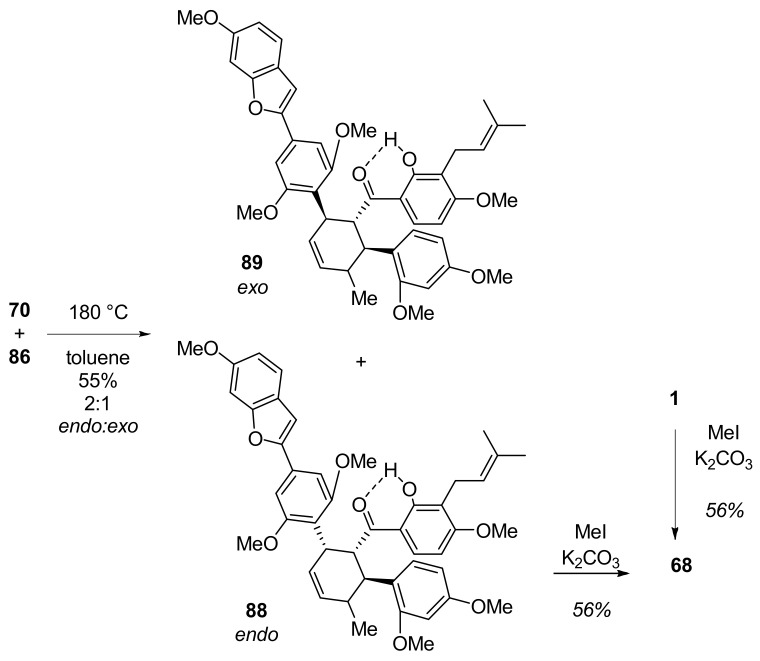
Diels–Alder reaction between **70** and chalcone **86** towards chalcomoracin heptamethyl ether (**68**).

**Figure 27 molecules-27-07580-f027:**
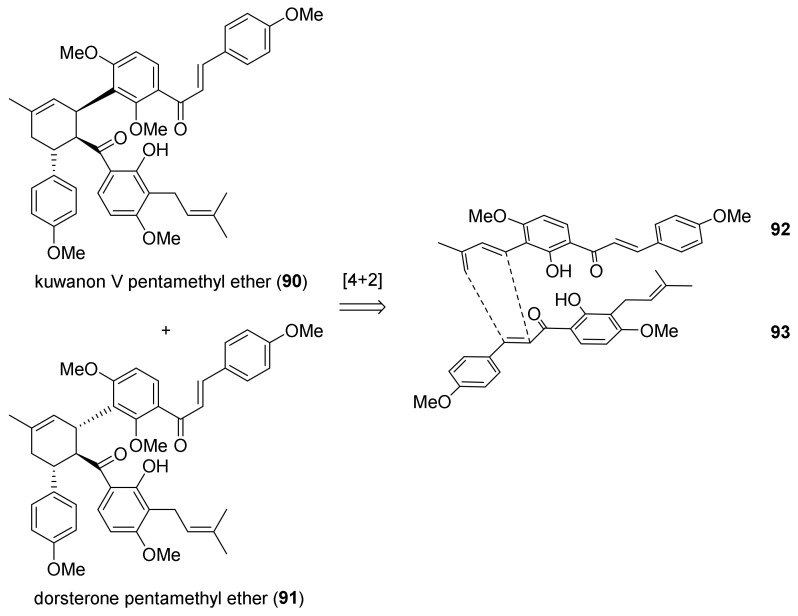
Retrosynthetic plan of kuwanon V and dorsterone methyl ethers (**90** and **91**).

**Figure 28 molecules-27-07580-f028:**
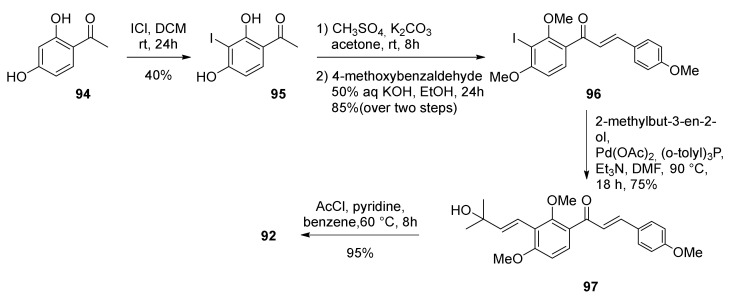
Synthesis of chalcone diene **92**.

**Figure 29 molecules-27-07580-f029:**
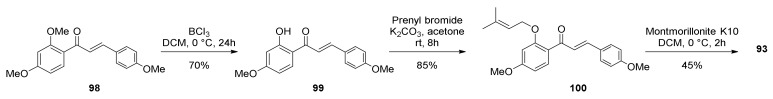
Synthesis of prenylated chalcone dienophile **93**.

**Figure 30 molecules-27-07580-f030:**
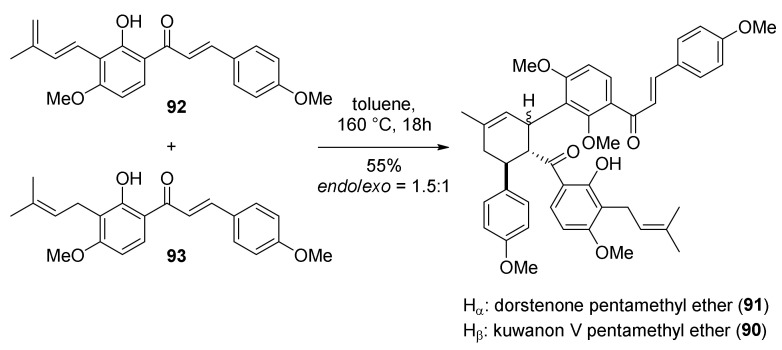
Synthesis of kuwanon V and dorsterone pentamethyl ethers (**90** and **91**).

**Figure 31 molecules-27-07580-f031:**
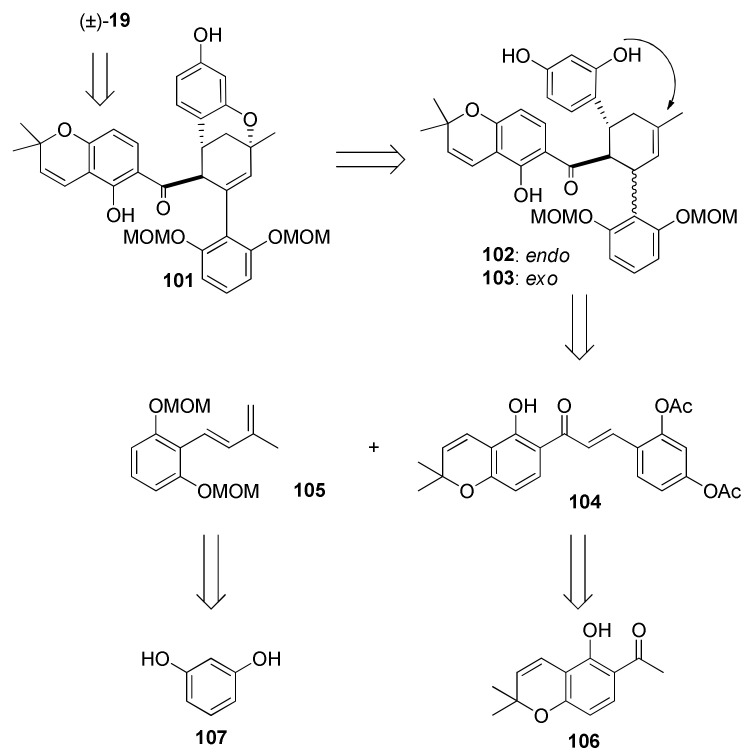
Retrosynthetic approach towards sorocenol B (**19**).

**Figure 32 molecules-27-07580-f032:**
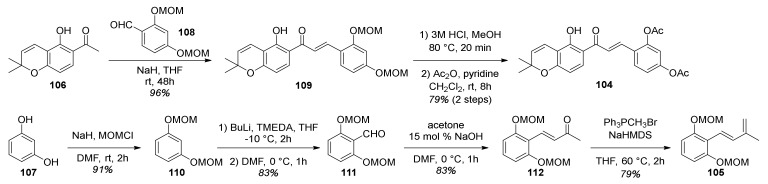
Preparation of diene **105** and chalcone **104**.

**Figure 33 molecules-27-07580-f033:**
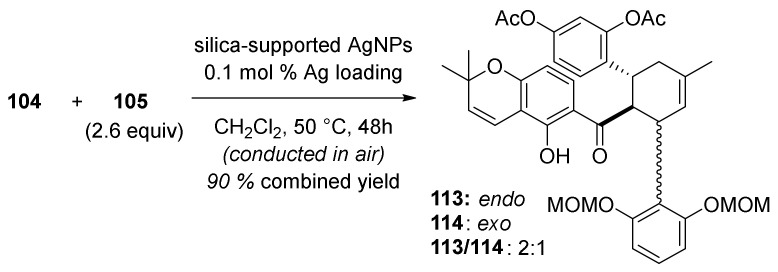
Diels–Alder cycloaddition between **104** and **105** catalysed by AgNPs.

**Figure 34 molecules-27-07580-f034:**
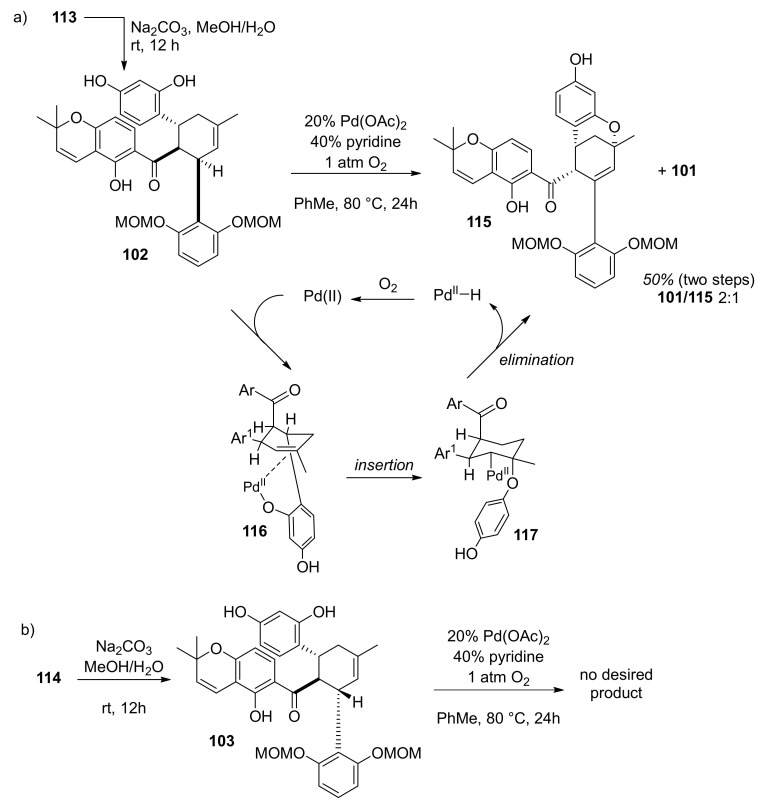
Pd(II)-catalysed oxidative cyclization.

**Figure 35 molecules-27-07580-f035:**
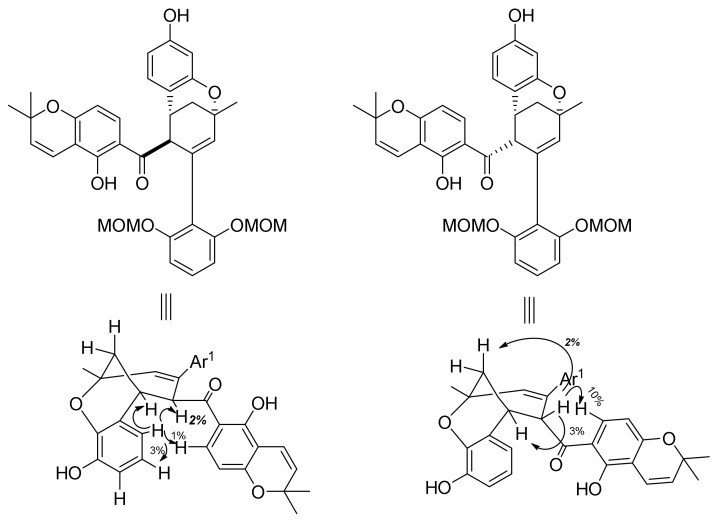
Key NOE’s leading to relative stereochemistry assignments of **101** and **115**.

**Figure 36 molecules-27-07580-f036:**
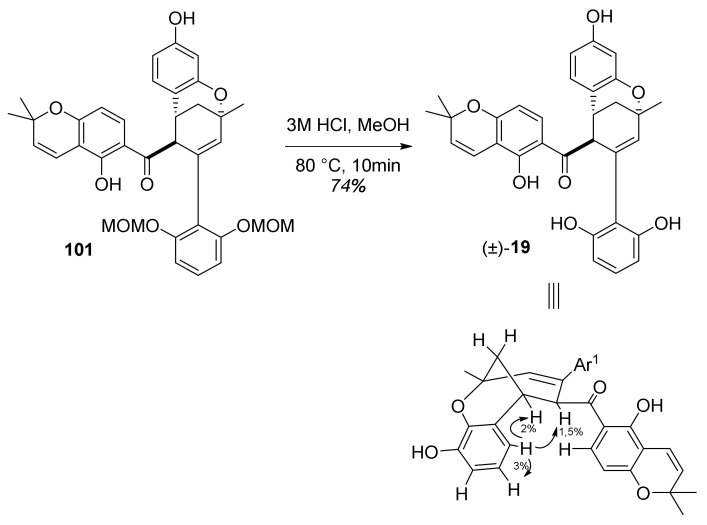
MOM-deprotection, last step yielding (±)-sorocenol B (**19**).

**Figure 37 molecules-27-07580-f037:**
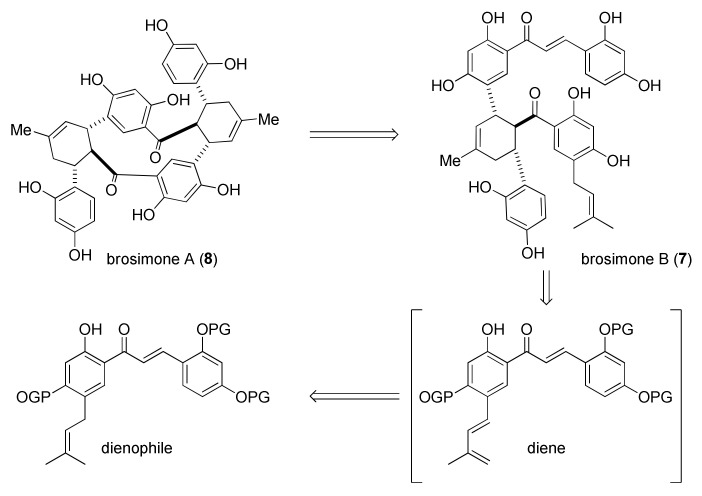
Retrosynthesis of brosimone A, derived from brosimone B.

**Figure 38 molecules-27-07580-f038:**
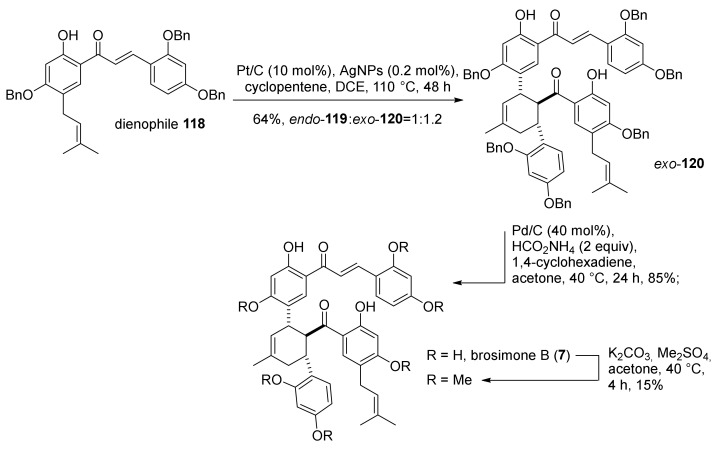
Dehydrogentive Diels–Alder cycloaddition towards brosimone B (**7**). Bn = benzyl.

**Figure 39 molecules-27-07580-f039:**
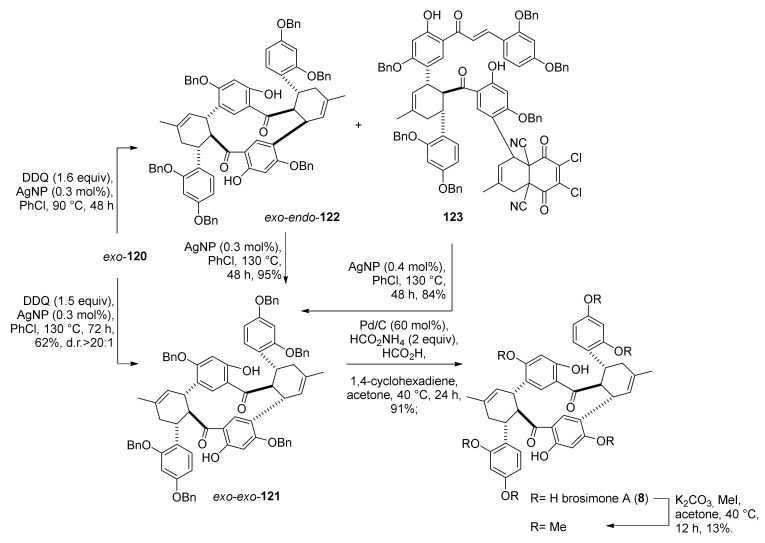
DHDA toward brosimone A derivative exo-exo-**121**.

**Figure 40 molecules-27-07580-f040:**
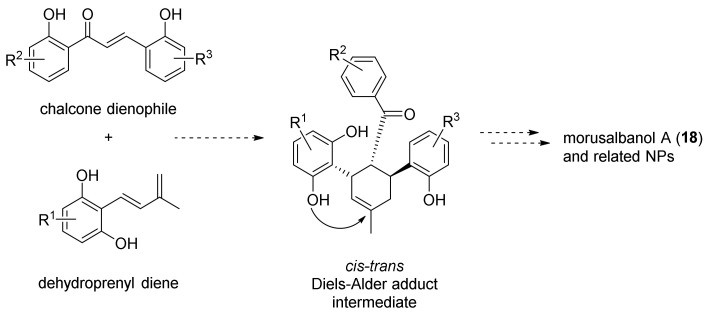
Biogenesis of morusalbanol A (**18**) and related natural products.

**Figure 41 molecules-27-07580-f041:**
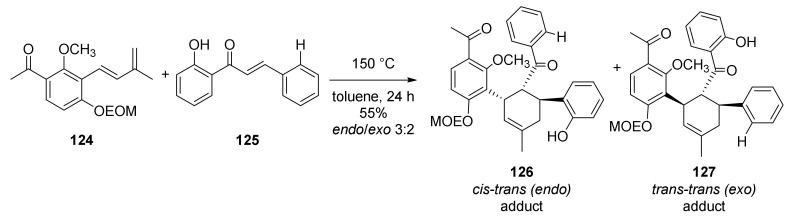
Development of initial methodology; best results achieved with model diene **124** and dienophile **125**.

**Figure 42 molecules-27-07580-f042:**
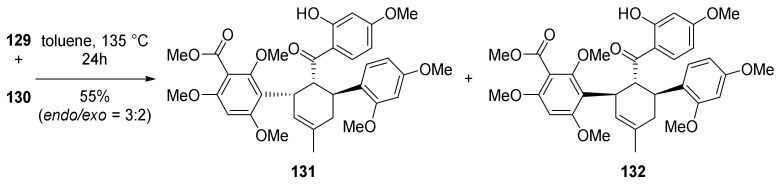
Thermal Diels–Alder reaction of diene **129** and dienophile **130**.

**Figure 43 molecules-27-07580-f043:**
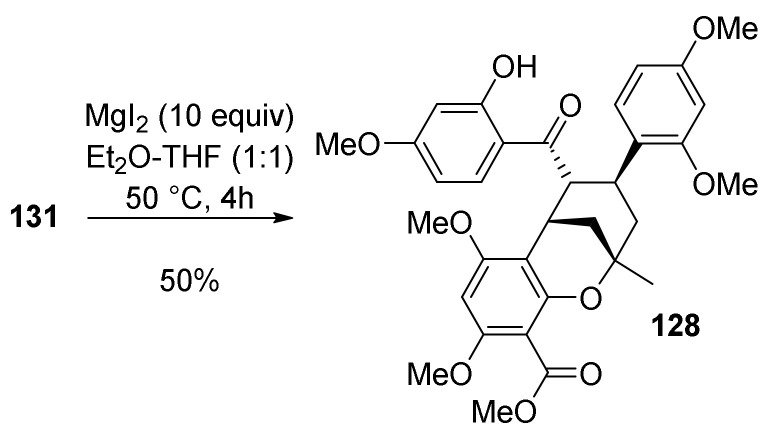
Synthesis of (±)-morusalbanol A pentamethyl ether (**128**).

**Figure 44 molecules-27-07580-f044:**
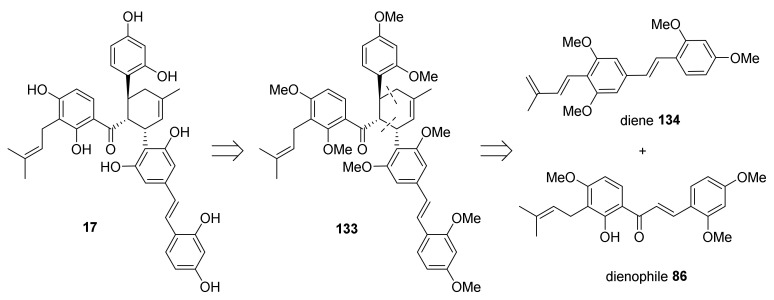
Retrosynthetic approach to kuwanol E (**17**).

**Figure 45 molecules-27-07580-f045:**
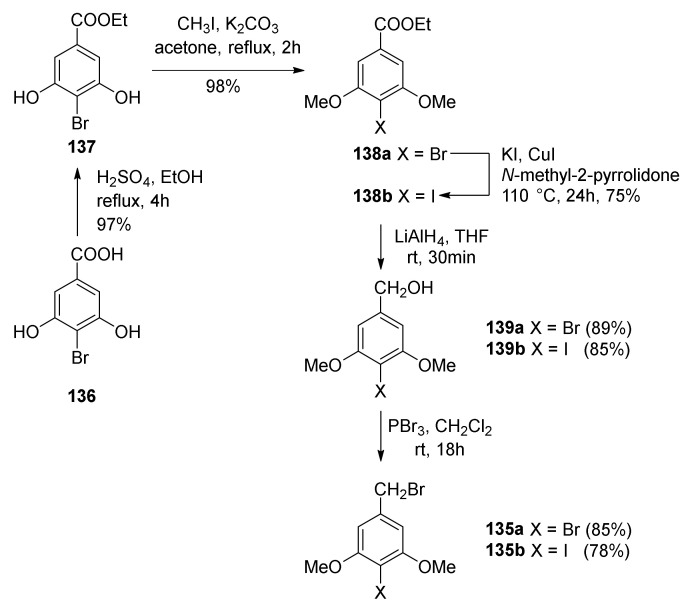
Synthesis of benzyl bromides **135a** and **135b**.

**Figure 46 molecules-27-07580-f046:**
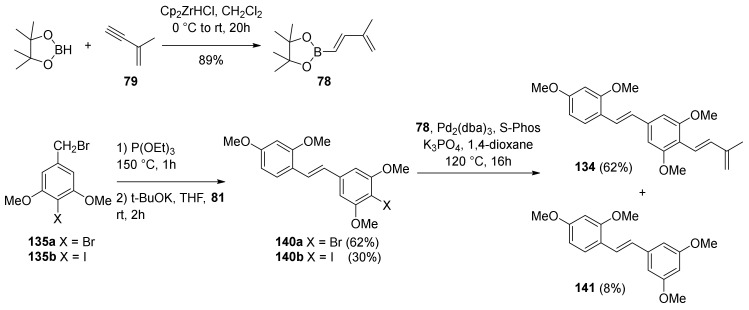
Synthesis of diene **134**.

**Figure 47 molecules-27-07580-f047:**
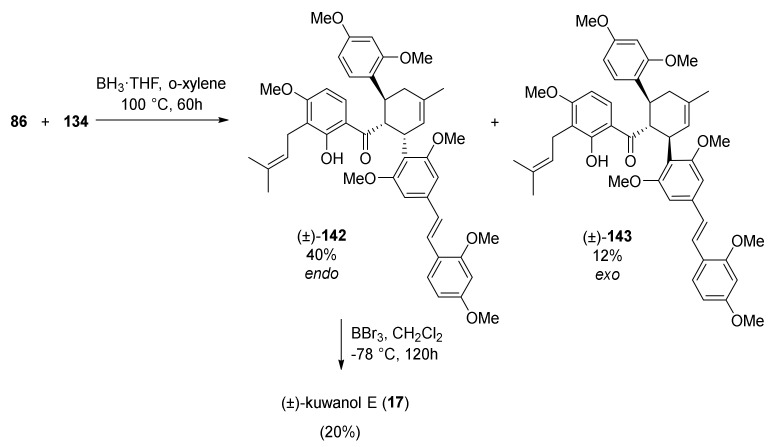
Synthesis of (±)-kuwanol E (**17**).

**Figure 48 molecules-27-07580-f048:**
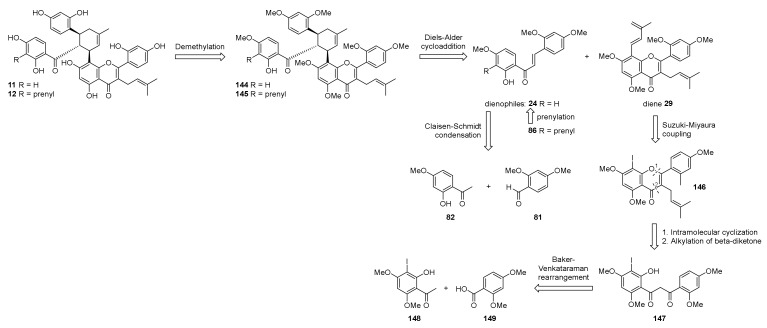
Retrosynthesis of kuwanons G and H (**11** and **12**).

**Figure 49 molecules-27-07580-f049:**
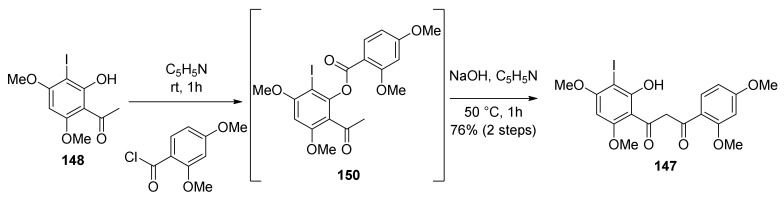
Preparation of iodo-β-diketone **147**.

**Figure 50 molecules-27-07580-f050:**
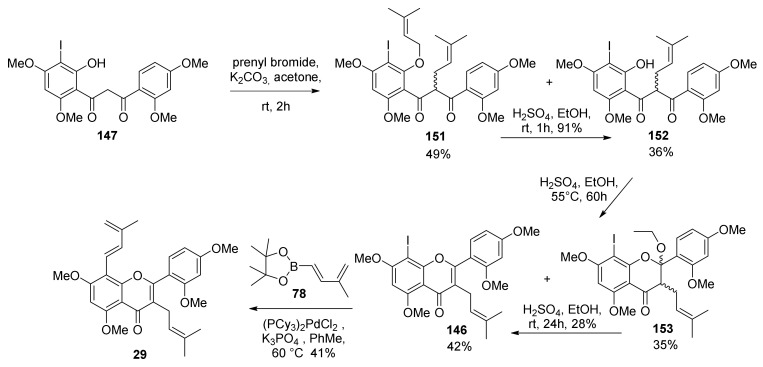
Synthesis of prenylated iodoflavone **194** and Suzuki–Miyaura cross coupling for the preparation of diene **29**.

**Figure 51 molecules-27-07580-f051:**
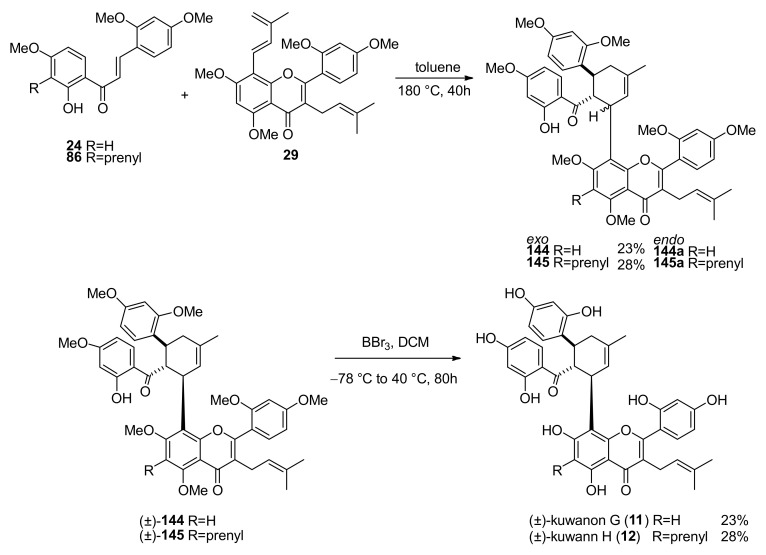
Synthesis of kuwanons G and H (**11** and **12**) by Diels–Alder reaction of dienophiles **24** or **86** and diene **29** and deprotection.

**Figure 52 molecules-27-07580-f052:**
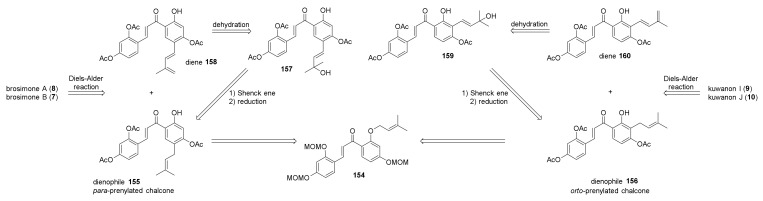
Retrosynthesis of brosimones A and B (**8** and **7**) and kuwanons I and J (**9** and **10**).

**Figure 53 molecules-27-07580-f053:**
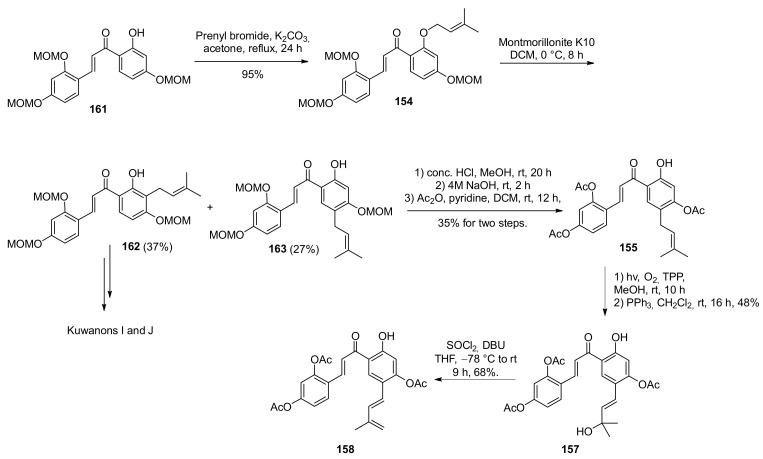
Preparation of chalcone diene **158**.

**Figure 54 molecules-27-07580-f054:**
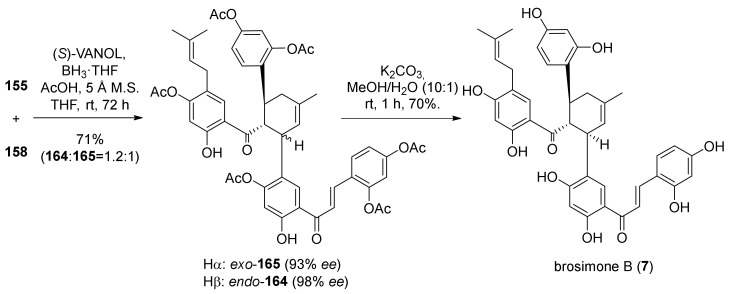
Enantioselective synthesis of brosimone B (**7**).

**Figure 55 molecules-27-07580-f055:**
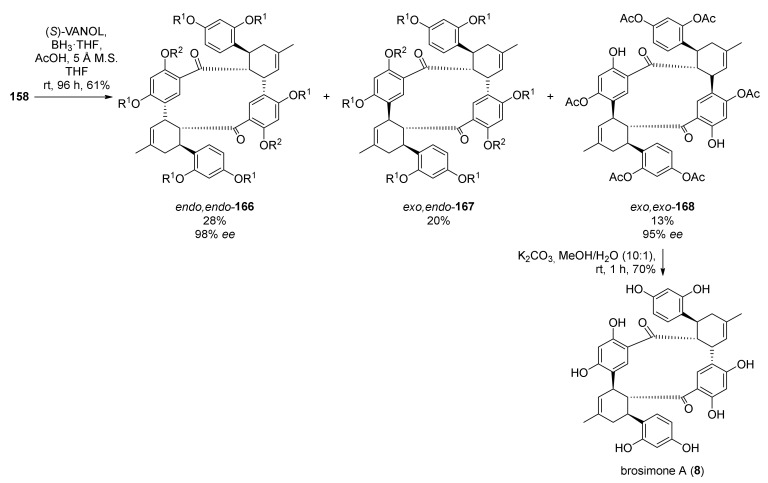
Enantioselective intramolecular Diels–Alder cycloaddition towards brosimone A.

**Figure 56 molecules-27-07580-f056:**
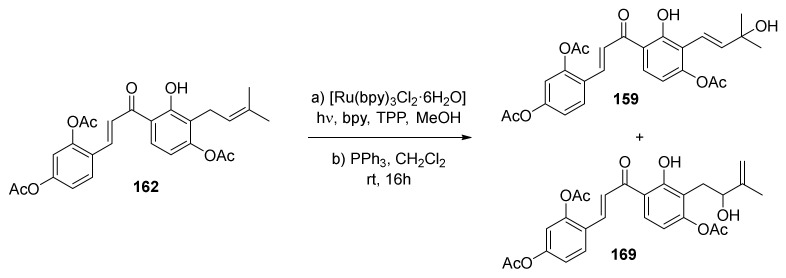
Schenck ene reaction towards tertiary alcohol **159**.

**Figure 57 molecules-27-07580-f057:**
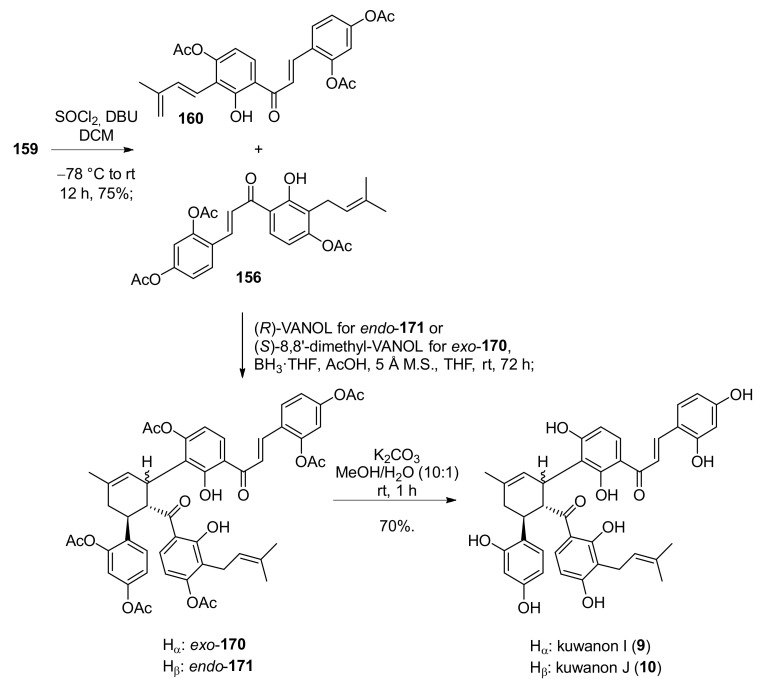
Enantioselective Diels–Alder cycloaddition towards kuwanons I and J (**9** and **10**).

**Figure 58 molecules-27-07580-f058:**
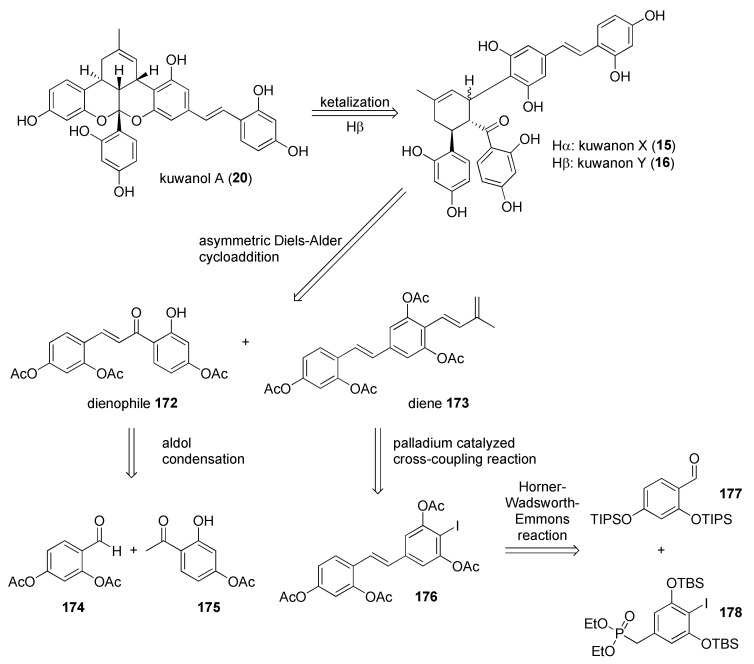
Retrosynthetic approach towards (−)-kuwanon X (**15**), (+)-kuwanon Y (**16**), and (+)-kuwanol A (**20**).

**Figure 59 molecules-27-07580-f059:**
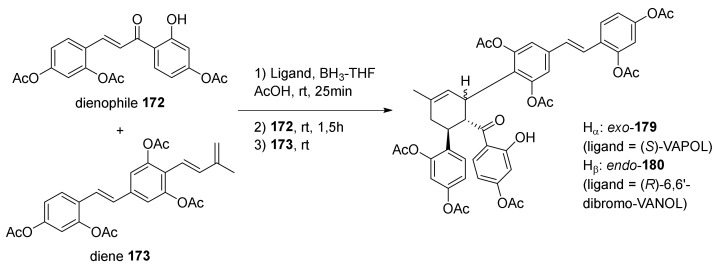
Asymmetric Diels−Alder cycloadditions from dienophile **172** and diene **173**.

**Figure 60 molecules-27-07580-f060:**
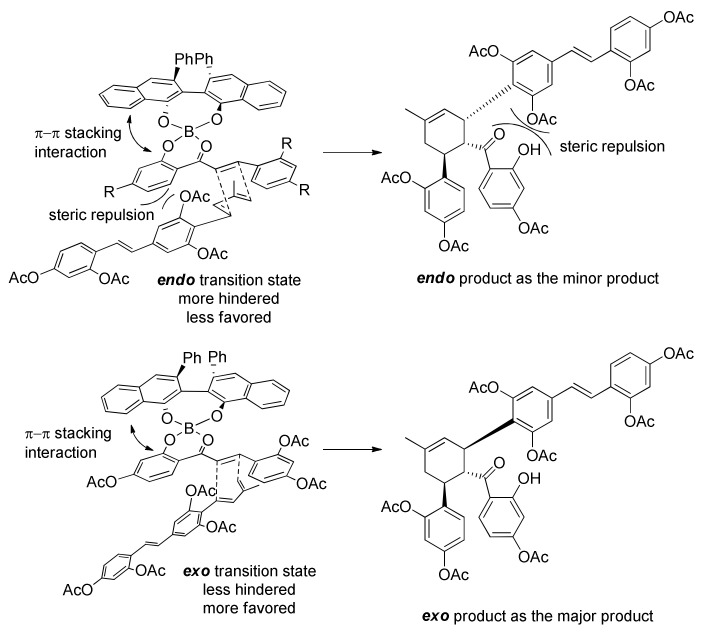
Transition states occurring during Diels–Alder cycloaddition between dienophile **172** and diene **173**, explaining the peculiar *exo*-selectivity.

**Figure 61 molecules-27-07580-f061:**
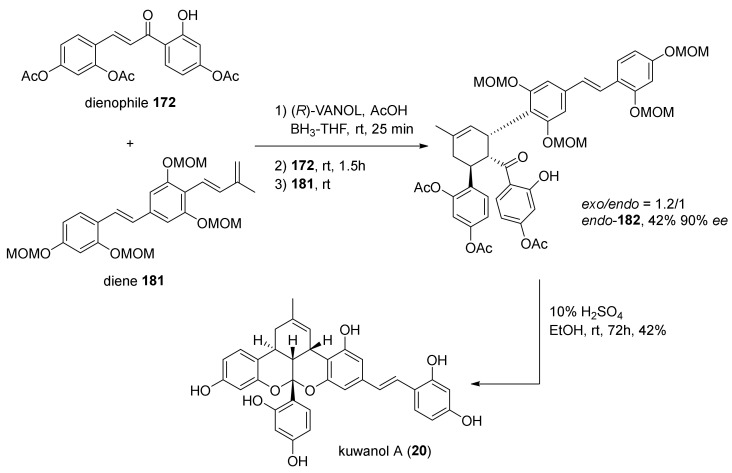
Synthesis of kuwanol A (**20**) from MOM-protected diene **173**.

**Figure 62 molecules-27-07580-f062:**
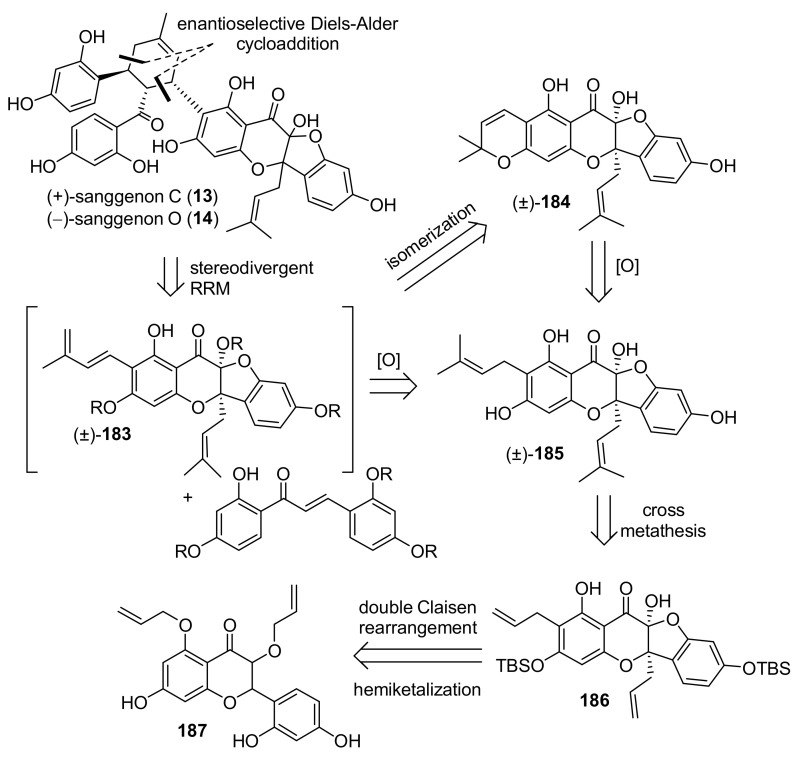
Retrosynthetic approach towards enantio-enriched sanggenons C and O, starting from sanggenon A (**184**) or sanggenol F (**185**).

**Figure 63 molecules-27-07580-f063:**
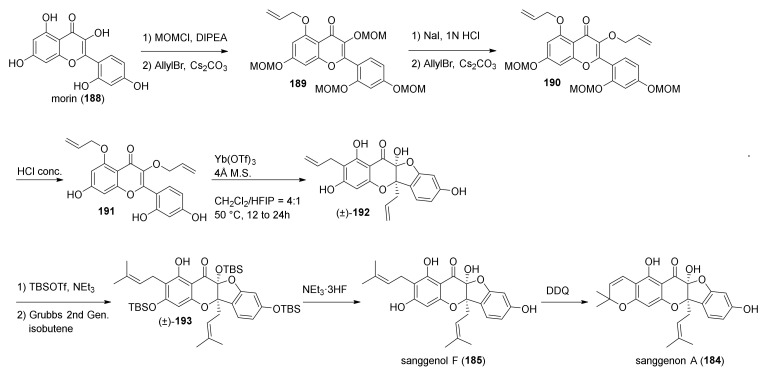
Synthesis of sanggenons A and F.

**Figure 64 molecules-27-07580-f064:**
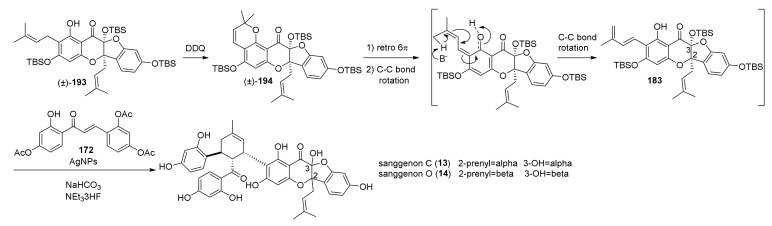
Synthesis of racemic sanggenons C and O (**13** and **14**).

**Figure 65 molecules-27-07580-f065:**
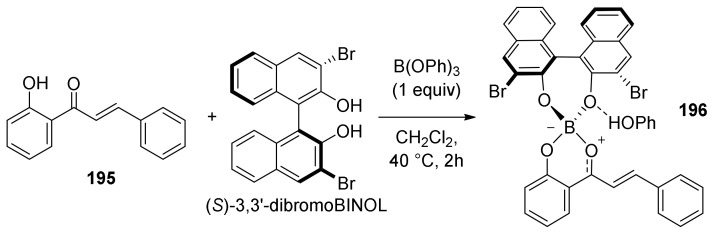
Borate chiral complex with model dienophile **195**.

**Figure 66 molecules-27-07580-f066:**
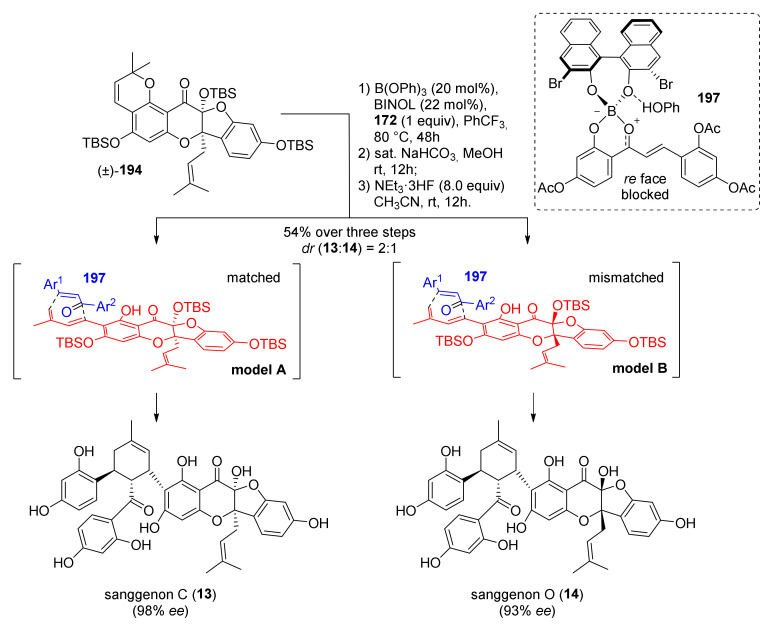
Stereoselective Diels–Alder cycloaddition towards sanggenons C and O.

**Table 1 molecules-27-07580-t001:** DAAs isolated from *M. alba* cell cultures and corresponding putative precursors.

DDAs Isolated from *M. alba* Cell Cultures	Prenylchalcone Precursor	Dehydroprenylphenol Precursor
kuwanon J	morachalcone A	dehydroprenylmorachalcone A
kuwanon Q	isobavachalcone	dehydroprenylmorachalcone A
kuwanon R	morachalcone A	dehydroprenylisobavachalcone
kuwanon V	isobavachalcone	dehydroprenylisobavachalcone
chalcomoracin	morachalcone A	dehydroprenylmoracin C
mulberrofuran E	isobavachalcone	dehydroprenylmoracin C

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
