# Peer review of "Synthesis, Biosynthesis, and Biological Activity of Diels–Alder Adducts from Morus Genus: An Update"

_molecules, 2022, doi:10.3390/molecules27217580_

Round 1

Reviewer 1 Report

This is an excellent review of the literature relating to natural products from Moraceous and related plants.  All reviews are increasingly important because of the rapid increase in the number of publications.  However, this is a particularly good review because it goes well beyond a simple compilation of compounds, and a great deal of thought has gone into the fundamental links between structures and the mechanisms of their biosynthesis.  Consequently, a very cohesive account is based on aspects of the Diels-Alder reaction, and adds considerably to an understanding of this field.  I recommend publication without any need for change.

Reviewer 2 Report

The study entitled "Diels-Alder adducts from Morus genus: synthesis, biosynthesis, and biological activity. An update" reports chemical and biological aspects related to the formation of Diels-Alder adducts in the Morus genus.

This is a properly described review that shows important findings on the biosynthesis of plant metabolites originating from [4+2] cycloaddition of different polyphenolic precursors. After the suggested corrections, I am in favor of publishing the manuscript.

 - In abstract, please change the term "Moraceae are plants that produce a" to "The plants of the Moraceae family are producers of a”;

- The authors have written the term “Morus root bark, also known as “Sang-Bai-Pi”. Please clarify the meaning. Morus is a genus of plants. So, does it mean that all the root barks of the species of this genus are known as “Sang-Bai-Pi”? If Morus is not a genre, then it should not be italicized in the article text.

- Please do not italicize the name of the botanical family, such as Moraceae. International standards for writing species names establish that only the name of the genus, species and subspecies must be written in italics.

- In line 41, "cis-trans isomers", cis should also be written in italics.

- On line 81, change the term "The Diels-Alder (DA)" to "The Diels-Alder (DA) reaction".

- In figure 2.2., according to the scheme in item d), please review the chemical structure of the product formed in "para"-like orientation.

- On line 175 write the meaning of CD.

- In figure 3.3. correct the word "coltures" by “cultures”.

- On line 220 change “Cudrania Tricuspidata” to “Cudrania tricuspidata”

- In figure 3.7 correct “M. Alba” to “M. alba”;

- M. notabilis must be written in italics in lines 328-329.

- In figure 5.18 the structure of brosimone A is wrong. Please review it.

- In line 593 Please correct "2-4-dimethoxybenzoic acid" to "2,4-dimethoxybenzoic acid".

- In lines 824-826 correct the concentration "MIC 0.78 Ag/ml" to "MIC 0.78 mug/ml".
